# Neural deficits in a mouse model of PACS1 syndrome are corrected with PACS1- or HDAC6-targeting therapy

Sabrina Villar-Pazos[1,6,8], Laurel Thomas[1,8], Yunhan Yang[1], Kun Chen[1,7], Jenea B. Lyles[1], Bradley J. Deitch[1], Joseph Ochaba[2], Karen Ling[2], Berit Powers[2], Sebastien Gingras[3], Holly B. Kordasiewicz[2], Melanie J. Grubisha[4,5], Yanhua H. Huang[4,5] & Gary Thomas[1]✉

PACS1 syndrome is a neurodevelopmental disorder (NDD) caused by a recurrent de novo missense mutation in *PACS1* (p.Arg203Trp (PACS1[R203W])). The mechanism by which PACS1[R203W] causes PACS1 syndrome is unknown, and no curative treatment is available. Here, we use patient cells and PACS1 syndrome mice to show that PACS1 (or PACS-1) is an HDAC6 effector and that the R203W substitution increases the PACS1/HDAC6 interaction, aberrantly potentiating deacetylase activity. Consequently, PACS1[R203W] reduces acetylation of α-tubulin and cortactin, causing the Golgi ribbon in hippocampal neurons and patient-derived neural progenitor cells (NPCs) to fragment and overpopulate dendrites, increasing their arborization. The dendrites, however, are beset with varicosities, diminished spine density, and fewer functional synapses, characteristic of NDDs. Treatment of PACS1 syndrome mice or patient NPCs with PACS1- or HDAC6-targeting antisense oligonucleotides, or HDAC6 inhibitors, restores neuronal structure and synaptic transmission in prefrontal cortex, suggesting that targeting PACS1[R203W]/HDAC6 may be an effective therapy for PACS1 syndrome.

Neurodevelopmental disorders (NDDs) frequently manifest as intellectual disability or autism spectrum disorder, with a combined incidence approaching 1:50 births[1]. The underlying genetic causes are often complex and allelically diverse[2]. Consequently, the biological pathways that become dysregulated to drive these disorders remain poorly understood, precluding informed therapeutic intervention[3]. Advances in next-generation sequencing methods identified recurrent de novo missense mutations in a handful of genes associated with NDDs[4]. Although these are rare mutations, they offer a unique opportunity to decipher the molecular pathways that cause complex diseases and may inform targeted therapeutic treatments. One such gene is *PACS1*, in which a single recurrent de novo missense mutation, c607C>T (p.Arg203Trp (PACS1[R203W])), was identified in patients who share an overlapping phenotype of developmental delay, intellectual disability, autism, seizures, and characteristic craniofacial dysmorphisms[5]. The detailed processes by which PACS1[R203W] causes PACS1 syndrome are unknown, and no curative treatment exists.

PACS1 is a multifunctional homeostatic regulator that is broadly expressed in all tissues, including the CNS, where it is most highly concentrated in neuronal centers[6–8]. In the cytoplasm, PACS1 partners with

[1]Department of Microbiology and Molecular Genetics, University of Pittsburgh School of Medicine, Pittsburgh, PA 15219, USA. [2]Ionis Pharmaceuticals, Carlsbad, CA, USA. [3]Department of Immunology, University of Pittsburgh School of Medicine, Pittsburgh, PA, USA. [4]Department of Psychiatry, University of Pittsburgh School of Medicine, Pittsburgh, PA, USA. [5]Translational Neuroscience Program, University of Pittsburgh School of Medicine, Pittsburgh, PA, USA. [6]Present address: Institute of Molecular Biotechnology of the Austrian Academy of Sciences (IMBA), Vienna Biocenter Campus (VBC), Vienna, Austria. [7]Present address: Department of Anesthesiology, Tongji Hospital, Tongji Medical College, Huazhong University of Science and Technology, Wuhan, China. [8]These authors contributed equally: Sabrina Villar-Pazos, Laurel Thomas. ✉e-mail: thomasg@pitt.edu

the WD40-repeat-containing protein, WDR37, to affect endoplasmic reticulum (ER) calcium signaling and partners with its paralogue, PACS2, to modulate retrograde trafficking of client proteins, including proteases, receptors, ion channels and pathogen proteins, between endosomes, the *trans*-Golgi Network (TGN) and the ER, and to the primary cilium[6,9,10]. In the nucleus, PACS1 and PACS2 interact with class I and class III histone deacetylases (HDACs) to support chromatin maintenance and DNA repair, and regulate metabolic gene expression[11–13].

Post-mortem studies suggest NDDs are frequently associated with dendritic abnormalities, both in their arbor complexity and spine density[14]. Dendrite arborization requires microtubule (MT)-dependent traffic to the developing branches and spines, which defines synaptic density, the size and scope of the receptive field, and the type of synaptic input[15]. MTs are organized by microtubule organizing centers (MTOCs), which tether their minus ends, and serve as roadways for dynein (retrograde)- and kinesin (anterograde)-dependent long-range transport activity. In proliferating cells, such as fibroblasts and neural progenitor cells (NPCs), the centrosome acts as the primary MTOC. In differentiating neurons, the centrosome is inactivated and repurposed (basal bodies) to assemble the primary cilium. This inactivation is carried out by the class IIb deacetylase HDAC6, which deacetylates α-tubulin to deploy Golgi elements to dendrites and has broad roles in cellular proteostasis control[16–19]. The deployed Golgi outposts support the elevated membrane flux needed to increase dendritic length, and possibly serve as secondary MTOCs that increase dendritic branching[20–23]. Accordingly, dysregulated HDAC6 activity causes abnormal dendritogenesis and is associated with several neuropsychiatric disorders[19].

Here we use patient fibroblasts, patient-derived NPCs, and mouse models to show that PACS1 is an in vivo effector of HDAC6 and that the R203W substitution represents a gain-of-function (GOF) mutation that increases the interaction between PACS1 and HDAC6 to aberrantly potentiate enzyme activity. Consequently, PACS1[R203W]/HDAC6 causes neuronal deficits in patient-derived cells and mouse models of PACS1 syndrome, including increased dendrite arborization together with reductions in spine density and synaptic strength. Treatment of the mice with antisense oligonucleotides (ASOs) that deplete HDAC6 or PACS1 restores both neuronal structure and communication, suggesting PACS1[R203W]/HDAC6 underlies PACS1 syndrome neuronal pathology and that this NDD may be treatable with targeted therapies.

## Results

### PACS1[R203W] disturbs Golgi positioning and microtubule acetylation

To begin to determine how PACS1[R203W] affects cellular function, we analyzed dermal fibroblasts isolated from PACS1 syndrome patients and their healthy parents (control, Fig. 1a). The patient cells were noticeably larger than their parental controls, irregularly shaped, and proliferated more slowly. Confocal analysis suggested PACS1[R203W] profoundly disturbed Golgi (Giantin) positioning and MT organization. In patient cells, the Golgi fragmented into dispersed mini-stacks, whereas in control cells the Golgi ribbon characteristically collected in the paranuclear region. The Golgi dispersal in patient cells was also observed by monitoring GM130, demonstrating these findings were independent of the Golgi-marker analyzed (Fig. S1a).

Consistent with their irregular shape, MTs in patient cells were disorganized and, unlike the parental control cells, failed to emanate from a single paranuclear MTOC. We, therefore, conducted an MT regrowth assay to ascertain the effect of PACS1[R203W] on MTOC integrity. Cells were treated with nocodazole for 10 hr to depolymerize MTs. The drug was then washed out and, after 3 min, the cells were fixed and processed for imaging to detect nascent EB1[+] MT asters emanating from MTOCs (pericentrin). Confocal analysis revealed patient cells contained multiple MTOCs, whereas control cells, as expected, contained a single paranuclear MTOC (Fig. S1b).

The limited number of available patient cell lines prevented us from conclusively determining that the R203W substitution is causal to the disturbed organellar positioning in the patient cells. We therefore analyzed heterozygous CRISPR/Cas9 PACS1[R203W] knock-in Hela cells (Hela[+/R203W]) and their isogenic WT control (Fig. 1b). Confocal analysis of the Hela-PACS1[+/R203W] cells showed a distended Golgi and disorganized MTs whereas the isogenic control Hela WT cells maintained the Golgi ribbon in the paranuclear region. Together, these findings suggest the R203W substitution is causal to both Golgi fragmentation and MT disorganization observed in the PACS1 syndrome patient cells.

Golgi positioning and MTOC organization are critically dependent upon the acetylation state of α-tubulin at Lys[40]. In fibroblasts, Ac-Lys[40]-α-tubulin stabilizes MTs, enabling the Golgi ribbon to collect over the paranuclear MTOC[24]. Conversely, α-tubulin deacetylation destabilizes MTs, causing dissolution of the MTOC and dispersal of Golgi ministacks. Consistent with this model, western blot analysis showed that the level of Ac-Lys[40]-α-tubulin is reduced in PACS1[R203W] patient cells (Fig. 1c). To determine whether the reduced level of Ac-α-tubulin contributes to the Golgi fragmentation, patient cells expressing K[40]Q-α-tubulin, an Ac-tubulin mimic, or non-acetylatable K[40]R-α-tubulin were analyzed by confocal microscopy (Fig. 1d). Golgi positioning was rescued by K[40]Q-α-tubulin but not K[40]R-α-tubulin, suggesting the reduced level of Ac-Lys[40]-α-tubulin in PACS1 syndrome patient cells causes both the Golgi fragmentation and disorganized MTs.

To test whether PACS1 has a physiologic role in regulating α-tubulin acetylation, we prepared embryonic fibroblasts (MEFs) from PACS1-deficient mice (Pacs1[Δ4bp/Δ4bp] (Pacs1[KO])) and their WT littermates (Fig. S1c and d). Western blot analysis showed Pacs1[KO] MEFs contain increased levels of Ac-Lys[40]-α-tubulin (Fig. 1e). Consistent with the increased level of Ac-α-tubulin in Pacs1[KO] MEFs, confocal imaging showed loss of *Pacs1* had no effect on Golgi positioning to the paranuclear region but as expected, caused the PACS1 client protein, furin, to mislocalize from the TGN to endosomes (Fig. 1f)[7]. The colocalization of endogenous PACS1 with a subpopulation of MTs further supports a role for PACS1 in regulating α-tubulin acetylation (Fig. S1e). Together, these studies suggest PACS1 is an in vivo regulator of α-tubulin acetylation and that the R203W substitution increases the PACS1-dependent deacetylation of α-tubulin to disturb Golgi positioning and MT organization.

### PACS1[R203W] binds HDAC6 and potentiates enzyme activity

Acetylation of α-tubulin is regulated by the class IIb lysine deacetylase HDAC6 and the class III enzyme SIRT2[25]. Co-immunoprecipitation (co-IP) studies showed that the R203W substitution increased the interaction between PACS1 and HDAC6 but had no effect on the interaction with SIRT2 (Fig. 2a and b). Next, PACS1[R203W] fibroblasts and their parental controls were treated with selective inhibitors of HDAC6 (tubacin, SW-100, and ACY-1215) or SIRT2 (AGK2) activity and processed for confocal microscopy. Only inhibition of HDAC6 rescued Golgi positioning (Fig. 2c and S2a). The importance of PACS1[R203W]/HDAC6 to the Golgi fragmentation was confirmed with siRNA knockdown. Depletion of either protein was sufficient to restore Golgi positioning in the PACS1[R203W] patient cells (Fig. S2b).

The physiologic interaction between PACS1 and HDAC6 was established by co-IP of the endogenous proteins in the patient and parental cell lines (Fig. S2c). Whereas the co-IP of the endogenous proteins did not show a significant effect of the R203W substitution on their interaction, confocal microscopy demonstrated a robust colocalization of endogenous PACS1/PACS1[R203W] and HDAC6 in p62 bodies only in the 159 patient fibroblasts. Image analysis revealed bodies positive for HDAC6 and PACS1/PACS1[R203W] were observed in 39% of patient cells (Pearson's coefficient = 0.55) but only 2% of parental control cells (Pearson's coefficient = 0.15), suggesting PACS1[R203W] also impacts HDAC6-mediated proteostasis pathways (Fig. 2d and see Discussion). We next investigated how PACS1[R203W] and HDAC6 interact

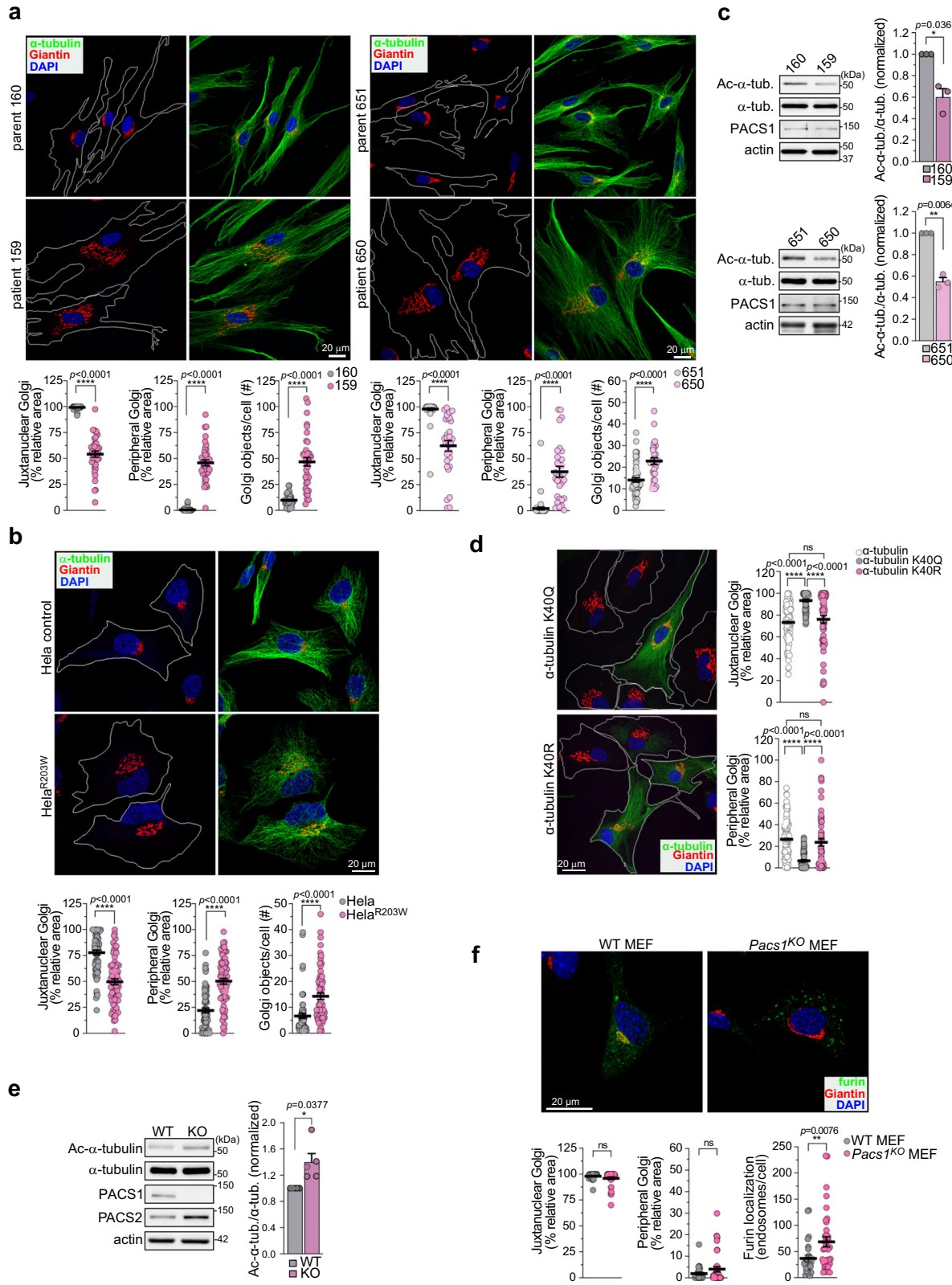

to affect enzyme activity. Of the 18 HDACs, only HDAC6 possesses two catalytic domains, CD1 and CD2, joined by a 39-amino acid linker region (Fig. 2e). CD2 accounts for the protein's physiological tubulin deacetylation activity whereas the biological function of CD1 is largely unknown[26]. Co-IP between PACS1 and a battery of HDAC6 deletion mutants showed that PACS1 interacts with the HDAC6 CD1+linker region. A reciprocal mapping strategy showed HDAC6 interacts with

multiple PACS1 domains. As expected, HDAC6 interacts with the PACS1 FBR, which harbors the disease-causing R203W substitution and binds numerous client proteins as well as trafficking adapters (Fig. 2f)[6,7]. In addition, HDAC6 interacts with the PACS1 CTR, a region of unknown function. However, the R203W substitution reduced the interaction between HDAC6 and a truncated PACS1 containing only the FBR (Fig. S2d). Next, we assessed the effect of PACS1[R203W] on HDAC6 activity.

**Fig. 1 | PACS1^R203W disturbs Golgi positioning and reduces α-tubulin acetylation.** **a** (Top) PACS1^R203W patient (159 and 650) and parent (160 and 651, respectively) fibroblasts were stained for Giantin (red), α-tubulin (green) and nuclei (DAPI). Cell perimeter (white). (Bottom) Quantification of Golgi fragmentation (Golgi objects/cell) and dispersal (percentage of total Golgi area within or beyond the juxta-nuclear/peripheral border (J/P) set 10 μm from the nuclear envelope (NE). Data are mean ± SEM (2-tailed *t*-test), *n* = 36 (160), 37 (159), 30 (650), or 51 (651) cells/group from three independent experiments. Scale bar, 20 μm. *p*-values shown in figure. **b** Hela^+/R203W cells and their isogenic WT control were stained for Giantin (red), α-tubulin (green), and nuclei (DAPI). (Bottom) Quantification of Golgi fragmentation and dispersal were quantified as in (a) except J/P was 3 μm from the NE. Data are mean ± SEM (2-tailed t-test), *n* = 65 cells/group from three independent experiments. Scale bar, 20 μm. *p*-values shown in figure. **c** Western blot of total α-tubulin and Ac-Lys^40-α-tubulin in patient (159 and 650) and parent (160 and 651) fibroblasts.

Data are normalized mean ± SEM (2-tailed *t*-test), *n* = 3 independent experiments, normalized individually to minimize inter-experimental variability. **d** Patient (159) cells expressing GFP-tagged K^40Q-α-tubulin or K^40R-α-tubulin were stained for Giantin (red), GFP (green), and nuclei (DAPI). Golgi area in GFP-positive or adjacent control cells (white bars) was measured as in (a). Data are mean ± SEM (1-way ANOVA), *n* = 105 (control), 62 (K^40Q), or 57 (K^40R) cells/group from three independent experiments. Scale bar, 20 μm. *p*-values shown in figure. **e** Western blot of total α-tubulin and Ac-Lys^40-α-tubulin in WT and Pacs1^KO MEFs. Data are mean ± SEM (2-tailed *t*-test), *n* = 5 independent experiments, analyzed as in (**c**). **f** WT and Pacs1^KO MEFs expressing human furin were stained for Giantin (red), furin (green), and nuclei (DAPI). Golgi area measured as in (a) except J/P was 4 μm from the NE. Furin^+ endosomes >0.25 μm were quantified. Data are mean ± SEM (2-tailed *t*-test), *n* = 27 (WT) or 36 (Pacs1^KO) cells/group from three independent experiments. Scale bar, 20 μm. Source data are presented as Sources Data files.

HDAC6 was immunoisolated from cells expressing HDAC6 alone or together with PACS1 or PACS1^R203W, and equal amounts of the deacetylase were assayed for enzyme activity in vitro (Fig. 2g). PACS1 had no measurable effect on HDAC6 activity. By contrast, PACS1^R203W increased HDAC6 activity by 1.5 fold, consistent with the reduced acetylation of α-tubulin in patient cells. Together, these findings suggest that the R203W disease substitution increases the multi-domain interaction between full-length PACS1^R203W and HDAC6 to aberrantly potentiate deacetylase activity.

### Silencing HDAC6 corrects PACS1^R203W-induced neural deficits

In developing neurons, the Golgi ribbon disassembles into outposts, which migrate into the growing dendrites to support arborization[21–23]. Thus, the increased Golgi fragmentation caused by PACS1^R203W in patient fibroblasts (Fig. 1) led us to ask whether the R203W disease substitution alters neuronal structure in vivo. To test this possibility, we used CRISPR/Cas9 gene editing to knock-in Cre-inducible, HA-tagged human PACS1 or PACS1^R203W at the murine ROSA26 safe harbor locus[27], which allowed us to simultaneously monitor the effect of R203W on neuronal physiology and the subcellular localization of PACS1 (Fig. S3a). The resulting R26^P1 and R26^P1R203W lines were crossed with *Emx1^Cre* mice to induce cassette expression in excitatory pyramidal neurons of the hippocampus and cerebral cortex, which is a key site of autism/intellectual disability (ID) pathophysiology[28]. Western blot and immunohistochemical staining showed *Emx1^Cre* faithfully induced expression of HA-tagged PACS1 or PACS1^R203W at levels similar to endogenous PACS1 in the cortex and hippocampus (Fig. 3a and b). As premature activation of HDAC6 can interfere with radial migration of cortical neurons[29], staining of coronal sections for SATB2 (upper layer marker) and CTIP2 (layer 5 marker) suggested that *Emx1^Cre*-induced PACS1^R203W does not appear to disturb overall cortical layering (Fig. S3b).

A closer examination of the CA1 region in the *Emx1^Cre;R26^P1* and *Emx1^Cre;R26^P1R203W* mice suggested PACS1 concentrates in the *stratum pyramidale* more than PACS1^R203W does (Fig. 3c). As the two HA-tagged proteins are expressed at similar levels in brain (Fig. 3a), this finding suggested the R203W substitution may cause PACS1 to distribute more broadly throughout the neuron. To test this possibility, as well as the effect of R203W substitution on Golgi positioning and dendrite morphology, we prepared dissociated hippocampal neurons from post-natal day 0 (P0) R26^P1 and R26^P1R203W pups. At DIV5, the neurons were fixed and processed for confocal microscopy and Imaris image analysis (Fig. S3c, d). In the DIV5 R26^P1 neurons, the intact Golgi ribbon concentrated in the cell body together with PACS1 as observed in vivo, with only a few elements extending into the base of some of the neurites (Fig. S3c, d). By contrast, the neurites in DIV5 R26^P1R203W neurons appeared longer than those in R26^P1 neurons, and the Golgi fragmented into numerous outposts, which deployed together with PACS1^R203W throughout the length of every neurite and were frequently observed together in varicosities (see also Supplementary Movie 1).

We next evaluated the impact of PACS1^R203W/HDAC6 on dendrite arborization in vivo. Specifically, R26^P1 and R26^P1R203W P1 pups were co-injected with an *Hdac6*-specific antisense oligonucleotide (H6ASO) or a negative control ASO (nASO), together with Cre-inducible AAV-FLEX-tdTomato, to space fill only the PACS1- or PACS1^R203W-expressing neurons, (Fig. 4a). A single injection of the H6ASO effectively and durably depleted cerebral *Hdac6* mRNA and protein, and consequently increased the level of Ac-Lys^40-α-tubulin (Fig. S4a, b). Sholl analysis of the tdTomato^+ CA1 neurons at P18 showed that PACS1^R203W increased dendritic arbor complexity, both in length and the number of branch points (Fig. 4c). Importantly, the H6ASO reversed this overbranching, resulting in a dendritic arbor indistinguishable from R26^P1 mice. To determine if the increased dendrite complexity in the R26^P1R203W hippocampus was coupled to an increase in dendritic Golgi, we stained 50 μm sections prepared from the same specimens with anti-GM130 to monitor the fate of Golgi elements in the tdTomato^+ neurons. Consistent with our quantitative studies in the dissociated hippocampal neurons (Fig. S3c, d), confocal imaging suggested that R26^P1R203W neurons treated with nASO harbored Golgi elements that deployed into the tdTomato^+ apical dendrite (Fig. 4c). By contrast, treatment with H6ASO caused the Golgi to return to the cell body. As expected, neither ASO affected the localization of the Golgi to the cell body in the R26^P1 neurons.

To assess whether the effect of PACS1^R203W on neuronal structure in the mouse hippocampus informs the mechanism of the disease mutation in humans, we used iPSCs generated from paired patient and parent fibroblasts and differentiated them into NPC monolayers (Fig. S4b). Interestingly, on day 18 of the neural induction protocol, the PACS1^R203W NPCs precociously generated neurite-like processes that were not yet apparent in the parental NPCs (Fig. 4d). In addition, and similar to the R26^P1R203W pyramidal neurons, the neurites in the patient-derived NPCs contained Golgi elements that disseminated from the cell body into the neurite. A 24-hr treatment with the HDAC6 inhibitor tubacin reversed this effect, restoring Golgi localization to the paranuclear region. These findings suggest the effects of PACS1^R203W/HDAC6 on neuronal structure in mice recapitulate important aspects of the human disorder.

Next, we examined the tdTomato^+ secondary apical dendrites and found that PACS1^R203W markedly reduced spine density but also induced accumulation of varicosities, characteristic of ID and neurodegenerative disorders (Fig. 5a and see Discussion). The H6ASO restored spine density in R26^P1R203W neurons to levels observed in R26^P1 control neurons, suggesting the PACS1^R203W/HDAC6 complex functions in dendritic spines. In support of this possibility, confocal analysis of DIV18 hippocampal neurons revealed a pronounced co-localization of HA-tagged PACS1 or PACS1^R203W with the glutamatergic spine maker, PSD95 (Fig. 5b). In addition, endogenous PACS1 and HDAC6 co-fractionated with synaptosomes and co-localized in PSD95^+ dendritic spines (Fig. 5c and S5a), consistent with a report that the *Pacs1* mRNA belongs to a small subset of neuronal transcripts that localize to

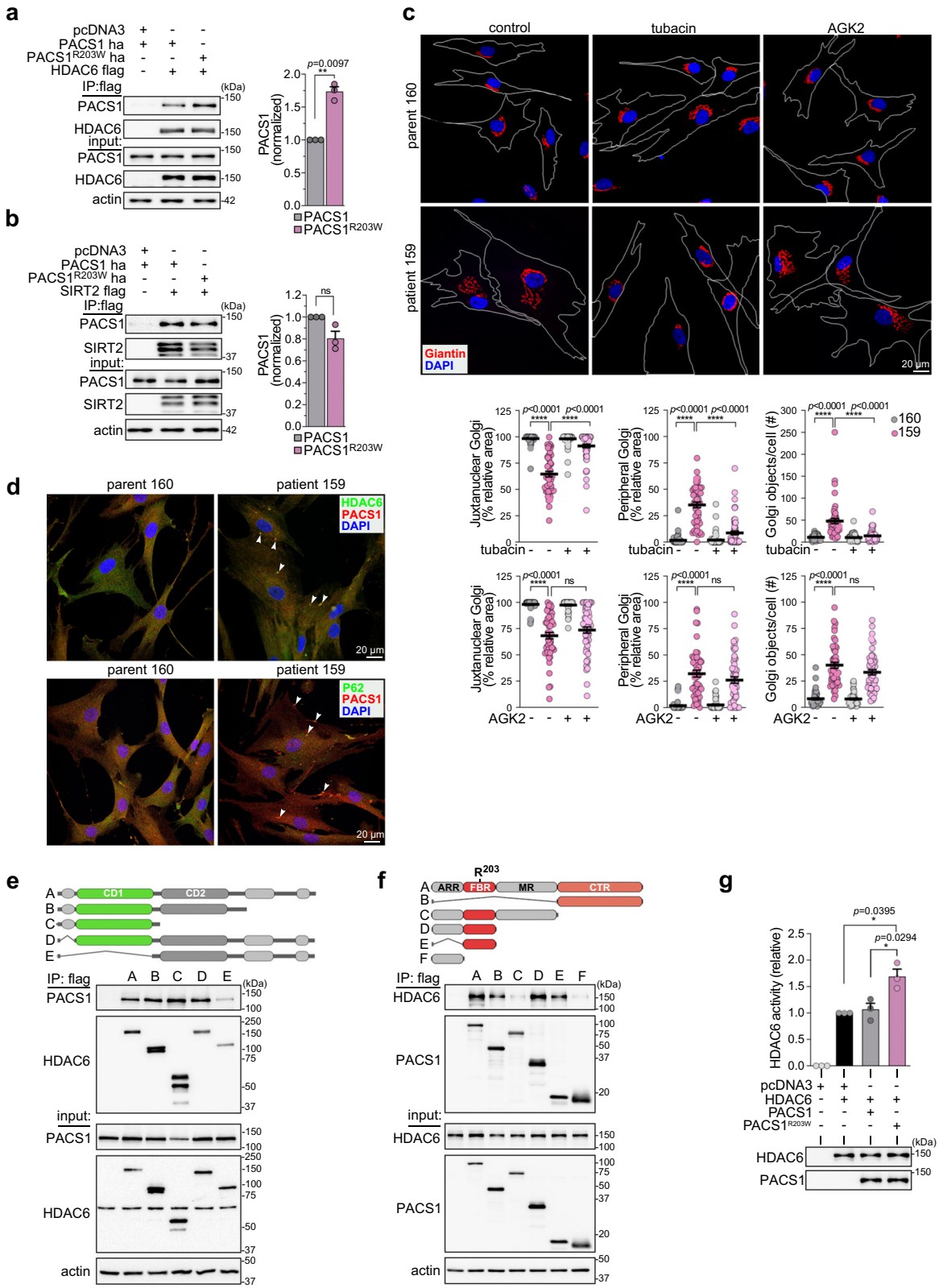

hippocampal synapses[30]. Notably, PACS1[R203W] reduced the level of acetylated cortactin (Ac-cortactin) in the cortex, which is a key HDAC6 substrate (Fig. 5d)[31]. As Ac-cortactin stabilizes the PSD95 scaffold that organizes cell-surface AMPA receptors (AMPARs)[32], these findings suggest PACS1[R203W] and HDAC6 traffic to dendritic spines where they trigger excessive deacetylation of cortactin to disturb synapse organization.

## HDAC6 ASOs correct synaptic transmission deficits in cortex

The lower spine density and reduced Ac-cortactin in R26[P1R203W] mice (Fig. 5a and d) suggested that the R203W substitution reduces the number of functional synapses. To test this possibility, brain slices were prepared from juvenile R26[P1] or R26[P1R203W] mice that had been treated on P1 with H6ASO or nASO (see Fig. 4a), and miniature post-synaptic current (mPSCs) were recorded from L2/3 pyramidal neurons

**Fig. 2 | PACS1^R203W^ interacts with HDAC6 to increase enzyme activity and disturb Golgi positioning.** **a**, **b** FLAG-tagged HDAC6 (**a**) or SIRT2 (**b**) were co-transfected with HA-tagged PACS1 or PACS1^R203W^ as indicated. FLAG-tagged proteins were immunoprecipitated and co-precipitating PACS1 proteins were detected by Western blot (anti-HA). Data are mean ± SEM (2-tailed *t*-test), *n* = 3 independent experiments, normalized individually to minimize inter-experimental variability. **c** (Top) Patient (159) and parent (160) cells treated with vehicle alone (DMSO, control), the HDAC6 inhibitor tubacin (5 µM, 4 hr), or the SIRT2 inhibitor AGK2 (10 µM, 16 hr). Cells were fixed and stained for Giantin (red) and nuclei (DAPI). Scale bar, 20 µm. (Bottom) Quantification of Golgi fragmentation and dispersal as described in 1a. Data are mean ± SEM (2-way ANOVA followed by Tukey *post hoc* test), *n* = 49 (160/veh), 47 (159/veh), 63 (160/tubacin), 58 (159/tubacin), and 44 (160/veh), 49 (159/veh), 41 (160/AGK2), 54 (159/AGK2) cells/condition, and three independent experiments. *p*-values shown in figure. **d** (top) PACS1^R203W^ patient (159) and healthy parent (160) fibroblasts were fixed and stained for endogenous PACS1 (red), HDAC6 (green), and nuclei (DAPI). (bottom) PACS1^R203W^ patient (159) and

healthy parent (160) fibroblasts were fixed and stained for endogenous PACS1 (red), p62 (green), and nuclei (DAPI). Arrowheads, co-localization (see Methods, *n* = 40 cells/group). Scale bar, 20 µm. **e**, **f** HCT116 cells co-expressing (**e**) PACS1-HA and the indicated FLAG-tagged HDAC6 constructs or (**f**) HDAC6-V5 and the indicated FLAG-tagged PACS1 constructs were harvested, FLAG-tagged proteins captured with M2 agarose and bound PACS1-HA (**e**) or HDAC6-V5 (**f**) detected by Western blot. PACS1 regions; ARR, atrophin-related region; FBR, furin(cargo) binding region; MR, middle region; CTR, C-terminal region. These experiments were repeated three times with similar results. **g** HCT116 cells expressing HDAC6-FLAG alone or together with HA-tagged PACS1 or PACS1^R203W^ were lysed, HDAC6-FLAG was captured with M2 agarose and bound proteins eluted with FLAG peptide. Equal amounts of HDAC6 protein from each sample (Alphaview, see Methods) were assayed for deacetylase activity (Fluor-de-Lys). Data are mean ± SEM (2-tailed *t*-test), *n* = 3 independent experiments, normalized individually to minimize inter-experimental variability. Source data are presented as Sources Data files.

within the medial prefrontal cortex (mPFC). PACS1^R203W^ markedly reduced AMPAR-mediated glutamatergic excitatory mPSC (mEPSC) amplitude and frequency but did not affect GABAergic inhibitory mPSCs (mIPSCs, Fig. 5e and S5b). Importantly, the H6ASO treatment in R26^P1R203W^ mice fully restored both mEPSC amplitude and frequency to levels observed in R26^P1^ mice, which were unaffected by HDAC6 depletion (Fig. 5e). These findings suggest PACS1^R203W^/HDAC6 disturbs basal glutamatergic transmission and, in agreement with others, that HDAC6 knockdown itself in mPFC has no deleterious effect on mPSCs[33].

### PACS1 ASOs correct synaptic transmission deficits in PACS1^R201W^ mice

The R26 knock-in strategy, while accelerating our ability to identify a key molecular pathway underpinning PACS1 syndrome, prevented testing whether PACS1^R203W^ itself can be safely and effectively targeted with ASOs, which would eliminate all molecular pathways disturbed by the disease mutation. We therefore mated heterozygous Pacs1^+/Δ4bp^ mice (Pacs1^HET^, see Fig. S1c), and determined that loss of *Pacs1* reduced survival by ~30% at weaning, and the surviving pups were smaller than their WT littermates but were viable and fertile (Fig. S6a and Table 1). In agreement with our finding that Pacs1^KO^ MEFs contain elevated Ac-Lys^40^-α-tubulin, western blot analysis demonstrated that loss of *Pacs1* also increased the levels of Ac-cortactin in the cortex (Fig. 1c and S6b).

Interestingly, the loss of *Pacs1* was coupled to an increased expression of its paralogue, *Pacs2*, both in the brain and in MEFs, suggesting PACS2 buffers an essential function of PACS1 (Fig. 1e and S1d). In support of this possibility, crossing *Pacs1^HET^;Pacs2^HET^* double heterozygotes failed to produce double knockout (DKO) pups (Table 2). Next, the effect of *Pacs1* loss on synaptic transmission was evaluated in L2/3 mPFC pyramidal neurons. Loss of *Pacs1* had no effect on mEPSC or mIPSC amplitude or frequency (Fig. S6c, d). Together, these findings suggest *Pacs1* loss does not significantly disturb excitatory or inhibitory synapses, supporting the development of a PACS1 ASO strategy for patient therapy. They further suggest that PACS1 is an in vivo HDAC6 modulator and that PACS2 compensates, to some extent, for the loss of PACS1.

We found that a standard germline knock-in of R201W (equivalent to human R203W) caused embryonic lethality. Therefore, we adapted a modified gene trap strategy to conditionally express the knocked-in mutation from the endogenous *Pacs1* locus (Fig. S7a). We also generated *Pacs1* WT mice harboring the gene trap while retaining Arg^201^ (Pacs1^M/+^), to control for any effects due to the inserted megamer. The gene-trapped lines were crossed with *Emx1^Cre^* mice, permitting a direct comparison with the R26 lines, and which correctly removed the traps (Fig. S7a). Next, the effect of the R201W mutation on basal synaptic transmission was evaluated in L2/3 mPFC pyramidal neurons from the *Emx1^Cre^*-induced Pacs1^R201W/+^ and Pacs1^M/+^ control mice. PACS1^R201W^

reduced mEPSC amplitude and frequency but had no effect on mIPSCs, similar to R26^P1R203W^ mice (Fig. 6a and S7b). Treatment of P1 pups with a *Pacs1*-specific ASO (P1ASO), which effectively and durably depletes PACS1 protein and mRNA (Fig. 6b and S7c), restored mEPSCs in Pacs1^R201W/+^ mice to levels observed in the matched control Pacs1^M/+^ mice.

Recent studies suggest PACS1 interacts with PACS2 and WDR37 to form a novel modulatory hub, and that in lymphocytes, expression of PACS1 and WDR37 are coupled such that loss of PACS1 reduces WDR37 protein levels[34–36]. Consistent with this model, ASO depletion of *Pacs1* profoundly reduced WDR37 protein levels in brain (Fig. 6b and S7d). A similar suppression of WDR37 was observed in Pacs1^KO^ mice (Fig. S1d). The P1ASO also reduced HDAC6 protein levels but only in the Pacs1^R201W/+^ mice. Interestingly, PACS1^R201W^ suppressed PACS2 levels, which were restored by the P1ASO. qPCR analyses showed that the P1ASO had no effect on the mRNA levels of PACS2, HDAC6 or WDR37 in cortex, suggesting PACS1 and PACS1^R201W^ control their expression through a posttranscriptional mechanism (Fig. 6c). Considering *Pacs2* buffers the loss of *Pacs1* (Table 2), these findings suggest PACS1^R201W^ posttranscriptionally prevents induction of a compensatory role for PACS2. They further suggest the benefit achieved with the P1ASO on synaptic activity resulted, at least in part, from the coupled reduction of PACS1^R201W^ and HDAC6.

## Discussion

We report that PACS1 is an in vivo HDAC6 effector and that a pathogenic interaction between PACS1^R203W^ and HDAC6 underpins neuronal deficits associated with PACS1 syndrome. The R203W GOF disease substitution increases the multi-domain interaction between PACS1 and HDAC6, aberrantly potentiating its deacetylase activity and pirating its posttranscriptional regulation. Together, PACS1^R203W^ and HDAC6 increase dendrite complexity and dendritic varicosities but reduce spine density and the number of functional synapses (see Fig. 6d). Targeting either HDAC6 or PACS1/PACS1^R203W^ with ASOs prevents these deficits, supporting a pathogenic PACS1^R203W^/HDAC6 interaction that underlies PACS1 syndrome.

The PACS genes arrived late in evolution, first appearing in lower metazoans where the single PACS gene mediates endosomal traffic in neurons[37]. In vertebrates, the PACS gene duplicated, forming *PACS1* and *PACS2*, and the two gene products acquired new trafficking motifs and phosphorylation sites that have markedly expanded their roles in vivo, such that the DKO causes embryonic lethality ([6] and Table 2). One emerging role for the PACS proteins is to regulate HDACs[11–13]. Here we show the HDAC6 CD1 + linker region interacts with the PACS1 FBR (Fig. 2e and f), which contains R203, and also the CTR, a region of unknown function but defines the PACS-1 family (Pfam #PF10254). The increased interaction with HDAC6, however, requires full-length PACS1, suggesting the PACS1^R203W^/HDAC6 interaction involves

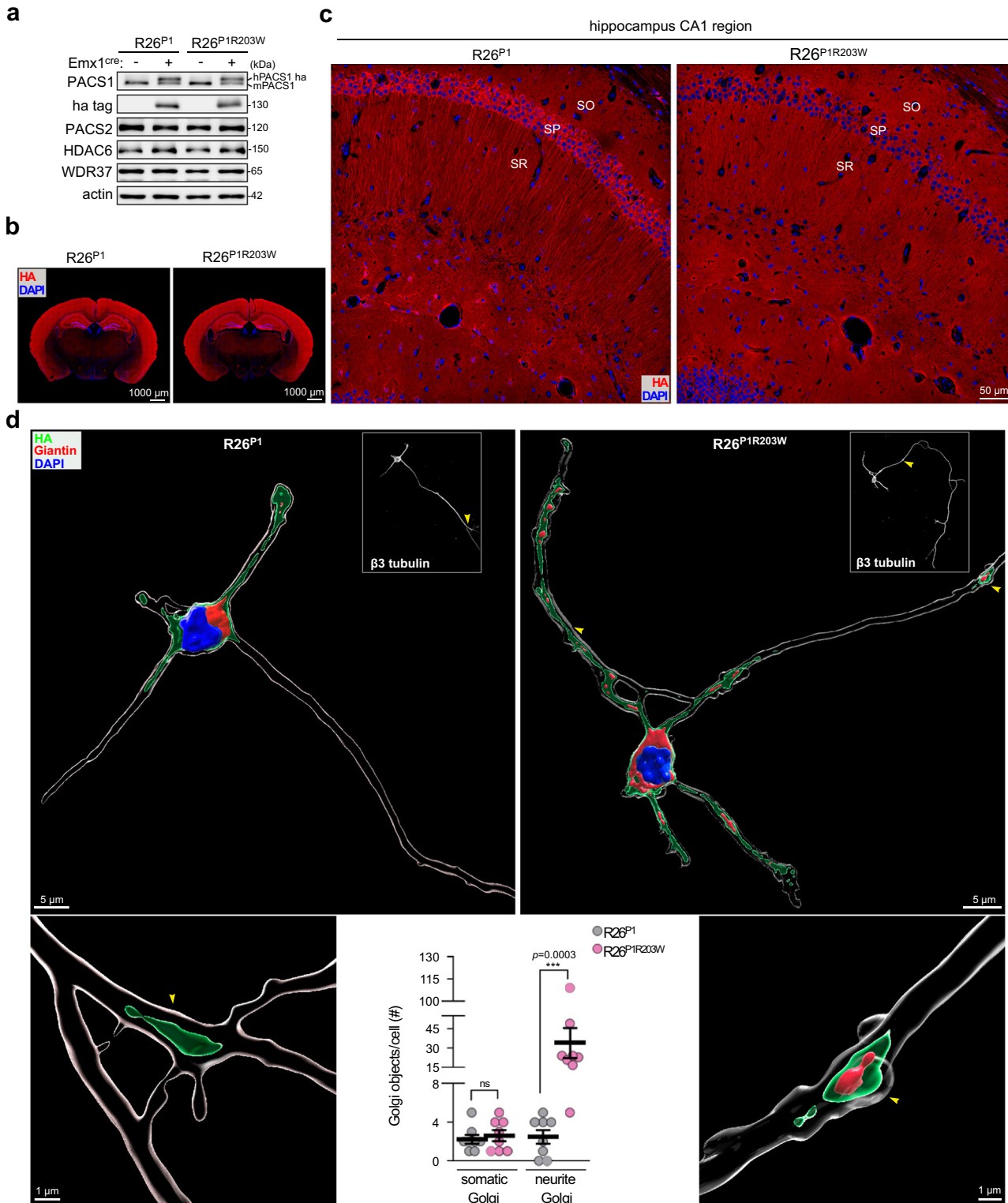

**Fig. 3 | PACS1$^{R203W}$ causes Golgi outposts to deploy into dendrites of hippocampal neurons. a, b** Western blots (**a**) and immunohistochemistry (IHC) of coronal brain sections (**b**) prepared from adult R26$^{P1}$ and R26$^{P1R203W}$ mice induced or not with *Emx1$^{Cre}$* to induce expression of HA-tagged PACS1 or PACS1$^{R203W}$ (red). Scale bar, 1000 μm. **c** Hippocampal CA1 region from specimens in (**b**). SO, *stratum oriens*; SP, *stratum pyramidale*; SR, *stratum radiatum*. Scale bar, 50 μm. **d** Dissociated *Emx1$^{Cre}$;R26$^{P1}$* or *Emx1$^{Cre}$;R26$^{P1R203W}$* hippocampal neurons (DIV5) were fixed and stained for β3-tubulin (white, insets), Giantin, and HA-tagged PACS1 or PACS1$^{R203W}$. High-resolution tiled images of entire cultured hippocampal neurons were captured using a confocal microscope (see Methods and Fig. S3c). (top) 3D surface

reconstructions (Imaris, see Methods) of the captured images depict Golgi elements (red), HA-tagged PACS1 or PACS1$^{R203W}$ (green), and nuclei (blue). Scale bar = 5 μm. (Bottom, left and right) Higher magnification of R26$^{P1}$ and R26$^{P1R203W}$ neurites (yellow arrowheads in the top panels) containing HA-tagged PACS1 or PACS1$^{R203W}$ and Golgi (R26$^{P1R203W}$ neurons). Scale bar = 1 μm. (Bottom middle) Quantification of deployed Golgi elements based on their subcellular localization. Somatic Golgi, within cell body; Neurite Golgi, Golgi deployed into developing neurites. Data mean ± SEM (2-tailed Mann−Whitney *U*-test), *n* = 8 cells/group from two independent experiments. See also Supplementary Movie 1. Source data are presented as Sources Data files.

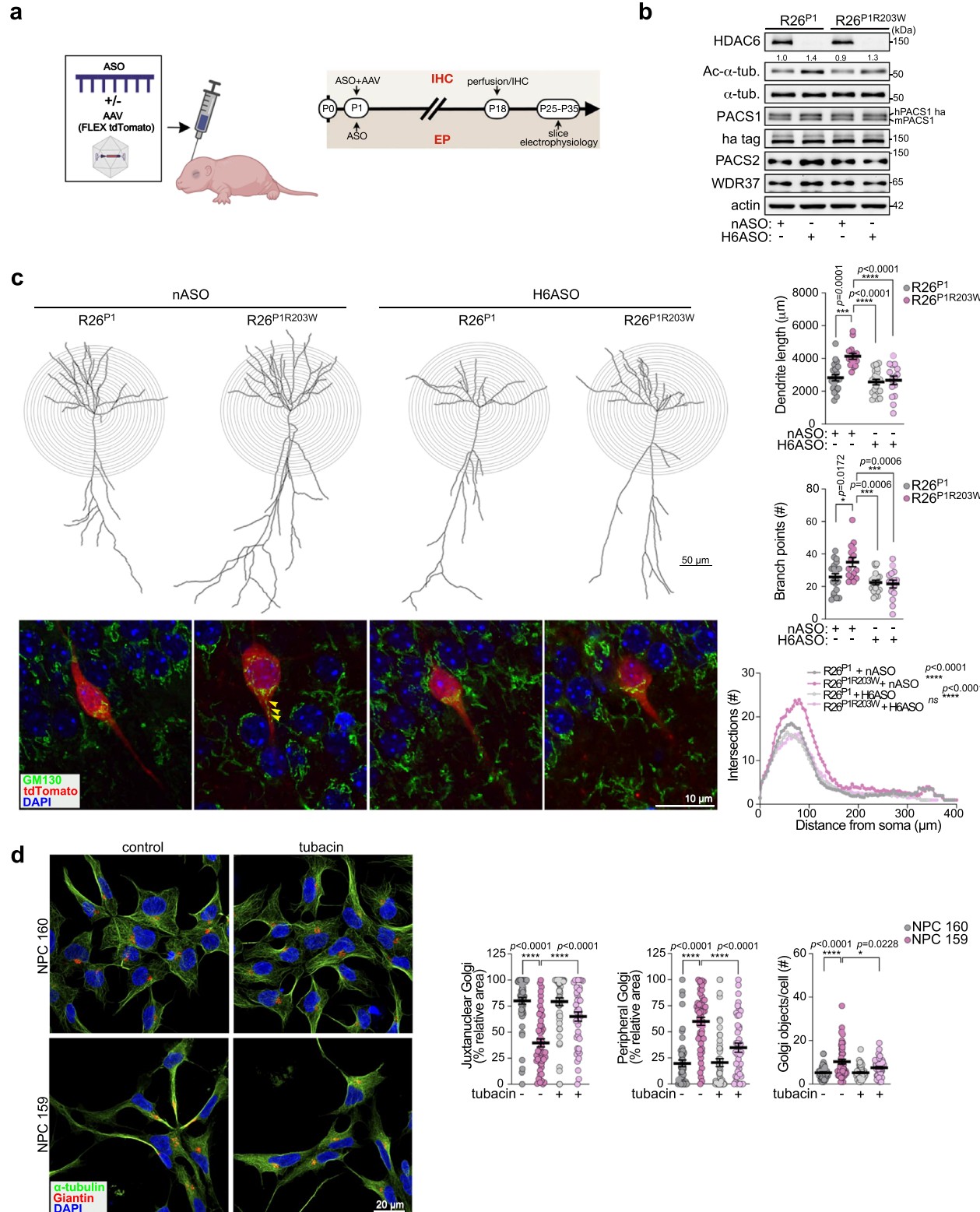

multiple surfaces on the two proteins, which combine to allosterically increase HDAC6 activity (Fig. 2g).

Genetic studies suggest that most HDAC6 interacting proteins, including p62 and CAMDI, negatively regulate enzyme activity[29,38,39]. Only two HDAC6-interacting proteins, Septin 7 and PACS1, increase deacetylase activity (ref. 40 and Fig. 2). Neither protein appears to activate HDAC6, suggesting they potentiate activity by targeting the deacetylase to its substrates (ref. 40, Figs. S1e and 5c). Speculatively,

we envision the spatially and temporally orchestrated actions of CAMDI, Septin 7 and PACS1 combine to modulate HDAC6-dependent dendrite formation in cortical neurons. Initially, CAMDI inhibits centrosomal HDAC6 activity, enabling the C-terminal ZnF domain to bind cdc20-APC. In turn, cdc-20 degrades the transcription factor Id1 to induce dendritogenic gene programs[17,29]. Subsequently, Septin 7 and PACS1 bind HDAC6 and target the deacetylase to its substrates, including α-tubulin and cortactin[40]. Thus, the intrinsic ability of

**Fig. 4 | HDAC6 ASO reverses the PACS1$^{R203W}$/HDAC6-dependent increased dendrite complexity in hippocampus. a** P1 *Emx1$^{Cre}$;R26$^{P1}$* and *Emx1$^{Cre}$;R26$^{P1R203W}$* pups were ICV-injected with nASO or H6ASO (40 µg) together or not with Cre-inducible AAV9-FLEX-tdTomato (1 × 10$^8$ vp) and then analyzed as depicted in the timeline (created with Biorender.com). **b** Cerebral cortices from postnatal *Emx1$^{Cre}$;R26$^{P1}$* and *Emx1$^{Cre}$;R26$^{P1R203W}$* mice treated with nASO or H6ASO were dissected and processed for Western blot with the indicated antisera. Numerical values depict the normalized signal intensity of Ac-α-tub bands determined with AlphaView (see Methods). **c** (Top) Representative dendritic arbor reconstructions of hippocampal CA1 pyramidal neurons from P18 *Emx1$^{Cre}$;R26$^{P1}$* and *Emx1$^{Cre}$;R26$^{P1R203W}$* mice treated with H6ASO or nASO. Scale bar, 50 µm. (Right) Dendrite length, number of branching points, and Sholl Analysis. Data are mean ± SEM (2-way ANOVA followed by Tukey *post hoc* test), *n* = 20 neurons/5 mice (R26$^{P1}$/nASO), 15 neurons/3 mice (R26$^{P1R203W}$/nASO), 20 neurons/5 mice (R26$^{P1}$/H6ASO), 15

neurons/3 mice (R26$^{P1R203W}$/H6ASO). *p*-values shown in figure. (Bottom) Brain sections from specimens in (**c**) were stained for Golgi (GM130), and Golgi positioning was assessed in tdTomato$^+$ CA1 pyramidal neurons. Z-stacks containing the whole tdTomato$^+$ dendritic arbor were trimmed to show focal planes containing the soma and proximal segments of the apical dendrite. Arrowheads disseminated Golgi elements in the primary dendrite. Scale bar, 10 µm. **d** (Left) Patient (159)- and parent (160)-derived NPCs were treated with vehicle (DMSO) or tubacin (5 µM, 24 h) and then fixed and stained for the Golgi-marker Giantin (red), α-tubulin (green) and nuclei (DAPI). Scale bar, 20 µm. (Right) Quantification of Golgi localized to the soma or disseminated down the neurite (Golgi objects/cell) and dispersal (J/P was 3 µm of the NE). Data are mean ± SEM (2-way ANOVA followed by Tukey *post hoc* test), *n* = 53 (160/veh), 49 (159/veh), 53 (160/tubacin), 43 (159/tubacin) cells/condition from three independent experiments. Scale bar, 20 µm. *p*-values shown in figure. Source data are presented as Sources Data files.

PACS1$^{R203W}$ to potentiate HDAC6 activity aberrantly reduces acetylation of α-tubulin and cortactin, leading to defects in neuron structure and function.

The reversible acetylation of α-tubulin is essential for MT nucleation at centrosomes (acetylated) and for driving the transition of the Golgi ribbon to dispersed mini-stacks (deacetylated), a step critical for dendritic growth and branching[41]. In flies, the Golgi does not present as a ribbon but is maintained as dispersed elements called "outposts", which efficiently sort into the growing dendrites[23]. The deployed Golgi elements localize to branch points where they support local secretory pathway trafficking needed by the growing arbor and may also function as secondary MTOCs[20,22,23]. Compared to flies, the characteristic Golgi ribbon in mammalian cells has been suggested to explain the relative infrequency of dendritic Golgi observed in mammalian neurons[20]. Nonetheless, in mammalian cortical and hippocampal pyramidal neurons, the Golgi and associated secretory cargo flux are polarized toward long primary dendrites, but not axons. Thus, the markedly elevated number of Golgi fragments dispersed into the dendrites in PACS1$^{R203W}$ neurons correlates with the increased neuronal arborization (Fig. 3d and S3c, and Supplementary Movie 1). The presence of Golgi elements along the presumptive axon in the PACS1$^{R203W}$ neurons was surprising and raises the possibility that one or more steps essential for excluding Golgi from axons may be compromised in PACS1 syndrome.

Importantly, the PACS1$^{R203W}$-induced fragmentation and dispersal of Golgi elements into neurites were observed in both R26$^{P1R203W}$ mice and in patient 159 NPCs (Figs. 3d, 4c, and 4d), suggesting this step informs the human disease. The ability of H6ASOs or small molecule inhibitors to reverse this phenotype in the mouse and human models further supports a role for PACS1$^{R203W}$/HDAC6 in PACS1 syndrome. The mechanism underlying the premature differentiation of the patient NPCs is unknown but is similar to that observed for patient-derived NPCs of other neurologic disorders, and which are also associated with aberrant neurite outgrowth and dysregulated HDAC6[42,43]. Although immunohistochemical staining suggests *Emx1$^{cre}$*-induced PACS1$^{R203W}$ does not grossly affect cortical layering (Fig. S3b), the altered phenotype of the PACS1$^{R203W}$ NPCs nonetheless raises the possibility that PACS1 syndrome may be associated with an imbalance in one or more neuronal or glial subpopulations[44].

The HDAC6-dependent dendritic overbranching in PACS1$^{R203W}$ neurons (Fig. 4c) is consistent with findings in other brain disorders, including ID, autism, and epilepsy, which also report structural abnormalities in dendrites and spines[14,45]. Autism is associated with dendritic overbranching and increased spine density[46]. By contrast, ID and epilepsy are associated with smaller dendrites and reduced spine density[14]. This reduced spine density in epilepsy may be a consequence of increased seizure-associated excitotoxicity[47]. Interestingly, our studies suggest PACS1$^{R203W}$ increases dendritic length but reduces spine density (Fig. 5a), raising the possibility that embryonic seizure activity may trigger a postnatal reduction in spine density. Our findings are similar to those reported for *Shank3B$^{-/-}$* mice, which also display

increased dendritic length but reduced spine density and reduced mEPSCs[48].

Acetylated cortactin organizes cell-surface AMPA receptors by stabilizing the submembranous PSD95 postsynaptic scaffold ([32] and see Fig. 6d). Thus, the reduced spine density in PACS1$^{R203W}$ neurons is consistent with the co-localization of PACS1$^{R203W}$ and HDAC6 in PSD95$^+$ dendritic spines (Figs. 5b and S5a), and with electrophysiology recordings demonstrating reduced mEPSC amplitude and frequency in both the R26$^{P1R203W}$ and Pacs1$^{R201W/+}$ mouse models (Figs. 5e and 6a). Our determination that ASO-mediated silencing of HDAC6 or PACS1 restores mEPSC amplitudes in PACS1$^{R203W}$ neurons is consistent with reports that deacetylated cortactin reduces clustering of the PSD95 postsynaptic scaffold and, in turn, suppresses mEPSC amplitudes[31,32]. Similarly, the ASO-mediated restoration of mEPSC frequency in the PACS1 syndrome mice is consistent with reports that HDAC6-dependent deacetylation of α-tubulin in the presynaptic neuron reduces mEPSC frequency[49]. The reduced mEPSCs but not mIPSCs suggest PACS1 syndrome may be associated with an E/I imbalance frequently observed in NDDs[50,51], and indicates a broad impact of PACS1$^{R203W}$/HDAC6 on neuronal communication. In addition to informing the number of functional synapses, mEPSCs stabilize the synaptic network, affect AP firing, and impact changes in local translation that are altered in ID and epilepsy[52,53]. Interestingly, PACS1 and WDR37 modulate calcium signaling by the ER, which is the cellular store that supplies calcium for action potential-independent mEPSCs[35,53]. However, an aberrant PACS1$^{R203W}$/WDR37 interaction alone is unlikely to be the underlying cause of PACS1 syndrome deficits we report since the reduced mEPSC phenotype observed in R26$^{P1R203W}$ mice was rescued by HDAC6 ASO, despite the presence of PACS1$^{R203W}$ and WDR37 (Fig. 4b and see 5e).

Recurrent missense mutations in PACS1, PACS2, and WDR37 each cause overlapping neurodevelopmental deficits[5,36,54], and proteomic and biochemical studies suggest the three proteins form a functional hub that links the cellular protein folding network (the chaperome) to the membrane traffic machinery[6,34]. Our studies extend this model and suggest PACS1 links the cellular chaperome to HDAC6. In addition to regulating tubulin acetylation, HDAC6 has key roles in stress granule dynamics and the aggresome-autophagy pathway, raising the possibility that PACS1 also affects these processes[55,56]. In support of this possibility, PACS1 and PACS1$^{R203W}$ control the posttranscriptional regulation of WDR37 and PACS2 (Fig. 6c). In addition, PACS1 syndrome patient cells contain PACS1(PACS1$^{R203W}$)$^+$/HDAC6$^+$ p62 bodies (Fig. 2d), and dendrites from PACS1$^{R203W}$ neurons contain numerous varicosities, suggesting PACS1 syndrome may involve a possible degenerative process frequently described in epilepsy, infantile neurobehavioral failure, and neurodegenerative disorders (Fig. 5a)[57].

Finally, our study suggests the tantalizing possibility that PACS1 syndrome may be treatable with HDAC6 inhibitors or RNA-based therapies that target PACS1 or HDAC6[58]. Both avenues correct neurologic deficits in mice and ASOs in particular are emerging as a powerful therapeutic platform for treating autism and other

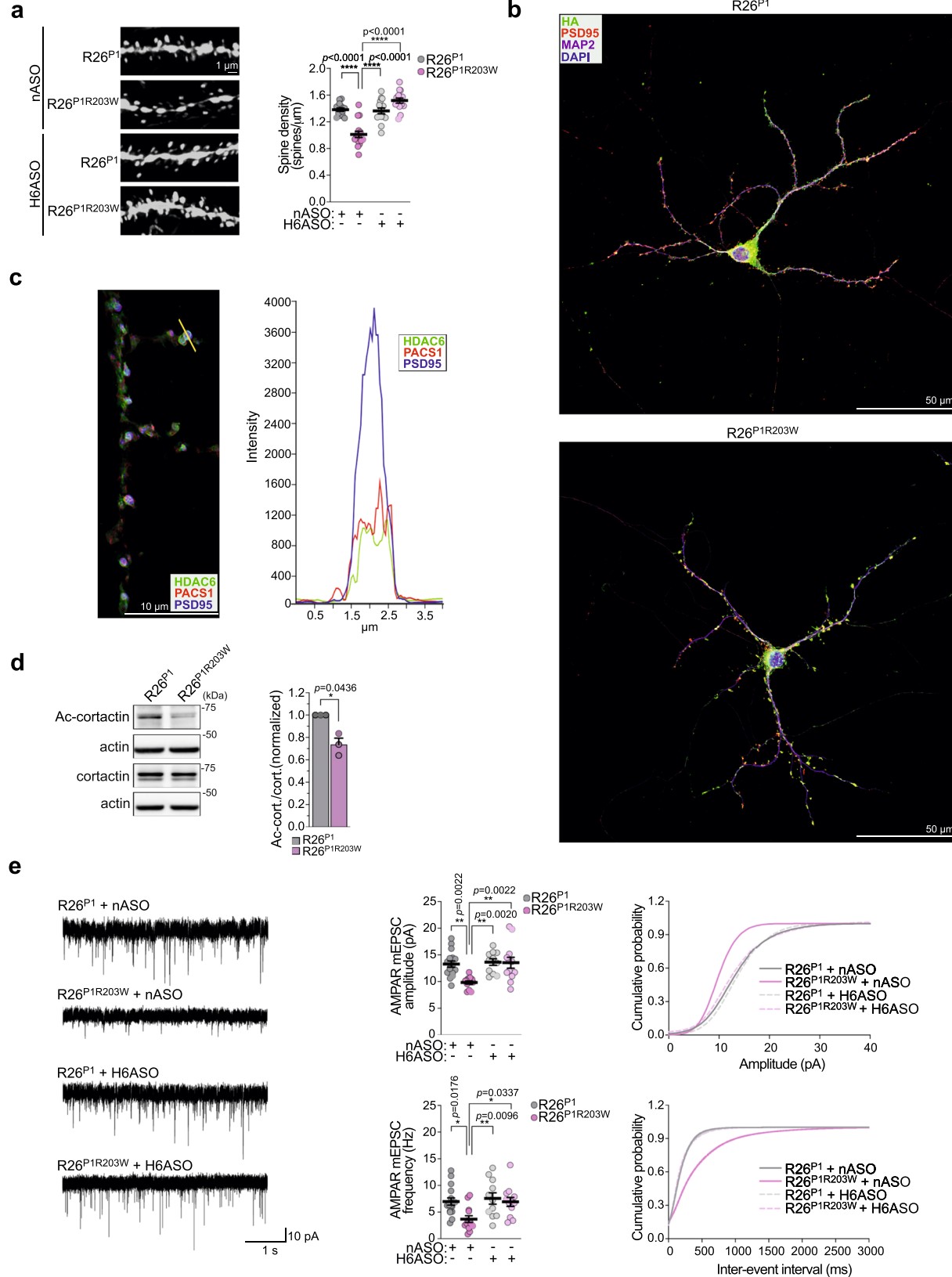

neurological diseases[58]. The viability and fertility of *Hdac6*[-/-] and *Pacs1*[-/-] mice, together with the lack of effect of loss of *Pacs1* on mEPSCs further supports these therapeutic avenues for the treatment of PACS1 syndrome. We anticipate similar strategies may be applied to combat the many other recently identified NDDs also caused by recurrent missense mutations.

## Methods

### Experimental animals

All animal protocols were approved by the University of Pittsburgh Institutional Animal Care and Use Committee and conducted in line with AVMA and ARRIVE guidelines. Male and female mice were used in these studies and housed in a temperature-controlled room and kept

**Fig. 5 | HDAC6 ASO rescues PACS1^{R203W}/HDAC6-induced deficits in spine density and excitatory synaptic communication. a** (Left) Representative images of secondary dendrites from the specimens in Fig. 4c with a spine density <1 SD away from the group mean. Scale bar, 1 μm. (Right) Data are mean ± SEM (2-way ANOVA followed by Tukey *post hoc* test), n = 14 neurons/59 dendrites (R26^{P1}/nASO), 16 neurons/55 dendrites (R26^{P1R203W}/nASO), 15 neurons/57 dendrites (R26^{P1}/H6ASO), 15 neurons/61 dendrites (R26^{P1R203W}/H6ASO), and 3 mice per condition. *p*-values shown in figure. **b** Dissociated hippocampal neurons isolated from P0 *Emx1^{Cre}*:R26^{P1} and *Emx1^{Cre}*;R26^{P1R203W} mice were processed for confocal imaging at DIV18 to detect MAP2 (magenta), HA-tagged PACS1 or PACS1^{R203W} (pseudocolored green), PSD95 (pseudocolored red) and DAPI (blue). Scale bar, 50 μm. **c** DIV18 neurons prepared from WT C57BL/6 mice were processed for confocal imaging to detect endogenous HDAC6 (green), PACS1 (red), and PSD95 (blue), and the fluorescent signal from the indicated spine (yellow line) was measured (Nikon Elements). Scale bar, 10 μm. **d** Western blot of Ac-cortactin, total cortactin, and actin in cortex isolated from *Emx1^{Cre}*:R26^{P1} or *Emx1^{Cre}*:R26^{P1R203W} mice. Data are mean ± SEM (2-tailed *t*-test), n = 3 independent experiments, normalized individually to minimize inter-experimental variability. **e** (Left) Representative whole-cell voltage-clamp recordings of mEPSCs from L2/3 pyramidal neurons in acute brain slices of juvenile *Emx1^{Cre}*;R26^{P1} and *Emx1^{Cre}*;R26^{P1R203W} mice injected (ICV) on P1 with 40 μg nASO or H6ASO (see Fig. 2e timeline). (Middle) Summary data of AMPAR mEPSC amplitude (top) and frequency (bottom). (Right) Cumulative probability distributions of AMPAR mEPSC amplitudes (top) and frequencies (bottom). n = 16 neurons/5 mice (R26^{P1}/nASO), 13 neurons/4 mice (R26^{P1R203W}/nASO), 11 neurons/3 mice (R26^{P1}/H6ASO), 12 neurons/3 mice (R26^{P1R203W}/H6ASO). Data are mean ± SEM (2-way ANOVA followed by Tukey *post hoc* test). Source data are presented as Sources Data files.

on a 12 hr light/dark cycle with food and water available ad libitum. C57BL/6 J mice (#:000664, RRID:IMSR_JAX:000664) and B6.129S2-*Emx1^{tm1(cre)Krj}*/J mice (#005628, RRID:IMSR_JAX:005628) were from Jackson Labs. Pacs2^{KO} mice were described[59]. All other engineered mice were generated by the Innovative Technologies Development and Mouse Embryo Services Cores in the Department of Immunology of the University of Pittsburgh, using CRISPR/Cas9 technology directly in C57BL/6 J zygotes. All single-guide RNAs were produced as described[60]. R26^{P1} (B6;*Gt(ROSA)26Sor^{tm1(CAG-LSL-Pacs1)Gath}*/J, ROSA26-LSL-PACS1-WT), and R26^{P1R203W} (B6;*Gt(ROSA)26Sor^{tm1(CAG-LSL-Pacs1-R203W)Gath}*/J, ROSA26-LSL-PACS1-R203W) lines were generated as described[27]; the HA-tagged cDNA for WT human PACS1 or PACS1^{R203W} were cloned into the pR26-GFP-Dest (a gift from Ralf Kuehn (Addgene plasmid # 74283)). Pronuclei of fertilized embryos (C57BL/6 J, The Jackson Laboratory), produced by natural mating, were microinjected with a mixture of 0.3 μM EnGen Cas9 protein (NEB M0646T), Rosa26-1 sgRNA (21 ng/μl ≈ 0.66 μM) and either the WT or the R203W targeting vector plasmid (10 ng/μl). The Pacs1^{Δ4bp/+} allele (Pacs1^{em1Gath}, Pacs1-4bp-del, Pacs1^{KO}) was generated by injecting C57BL/6 J zygotes with EnGen Cas9 protein and the guide Pacs1-e4-17Rev target sequence (5′-TACAA-GAACCGAACTATCTTGGG-3′, chr19:5,160,008-5,160,030, mouse GRCm38/mm10 assembly), which resulted in a 4 bp deletion leading to a frameshift null allele. Pacs1^{R201W/+} (Pacs1^{tm1Gath}, Pacs1-LSL-R201W) lines were generated using the Easi-CRISPR strategy[61]. Briefly, the zygotes were injected with 0.33 μM EnGen Cas9 protein, 21.23 ng/μl (~0.66 μM) sgRNA (Pacs1-i3-18rev, 5′-TAGCTGGACCTGAACCACCAAGG-3′, chr19:5,160,141-5,160,163, mouse GRCm38/mm10 assembly) and 10 ng/μl Pacs1-R201W-cKO Megamer (IDT). Pacs1^{M/+} (Pacs1^{tm2Gath}, Pacs1-LSL-R201) was identified in the process; the Lox-Stop-Lox cassette was inserted in the intron, but R201 remained unchanged, providing a negative control for the strategy used to generate the Pacs1^{R201W/+} knock-in mice. The sequence of the Pacs1-R201W-cKO-ssODN was: 5′-ttctctgaccaatcactcttacaaccaagagacgctaaagccactaacactcagtcctctagctg gacctgaaccaccaATAACTTCGTATAGCATACATTATACGAAGTTATTAG GGCGCAGTAGTCCAGGGTTTCCTTGATGATGTCATACTTATCCTGTC CCTTTTTTTTCCACAGCTCGCGGTTGAGGACAAACTCTTCGCGGTCTT TCCAGTcGATTACAAGGACGACGATGACAAGTAGATAACTTCGTATAG CATACATTATACGAAGTTATgaattcaggttctggttgctatcctggttaaccatttagt ctgttttcctcttcagtaccctcatttccttaagcgagatgccaacaaactgcagattatgttgcaa aggaggaagcggtacaagaacTgGacCatcttgggctataaaaccttagctgtgggactcatc aacatggcagaggtgagcagaacacaagtcttagactgcaggtct-3′. Oligonucleotides used for genotyping are presented in Table S1.

## ASOs and intracerebroventricular (ICV) injections

The MOE [2′-O-(2-Methoxyethyl)] gapmer ASOs used in this study were generated as described[62]. The negative ASO (nASO, 5′-CCTA-TAGGACTATCCAGGAA – 3) and mouse *Hdac6* ASO (H6ASO, 5′-GCCTACTCTTTCGCTGTC-3′) were previously reported[62], and the mouse *Pacs1* ASO (P1ASO, 5′-TCTCATTTTGACTATACCAT-3′) was screened and validated as described[63]. ICV injections were performed

on genotyped P1 pups as described[64]. Pups were cryo-anesthetized for 8 min, and the injection site was located ~0.25 mm lateral to the sagittal suture and 0.5–0.75 mm rostral to the neonatal coronary suture. ASOs diluted to 20 μg/μl in PBS (no calcium or magnesium) containing 0.01% Fast Green were drawn into a Hamilton syringe with a 32 g needle, positioned perpendicular to the skull surface, and impaled to a depth of 2 mm into the right hemisphere to deliver a single dose of 40 μg (2 μl injection) of ASO into the ventricle. After 10 s, the needle was slowly retracted to prevent backflow. The pups were gently warmed on a heating pad until fully recovered, returned to the nest, and monitored daily.

## Slice electrophysiology

Animals were anesthetized with isoflurane and then decapitated. Brains were quickly removed and placed in ice-cold NMDG-HEPES aCSF pH 7.3 (92 mM NMDG, 2.5 mM KCl, 1.25 mM NaH2PO4, 30 mM NaHCO3, 20 mM HEPES, 25 mM glucose, 2 mM thiourea, 5 mM Na-ascorbate, 3 mM Na-pyruvate, 0.5 mM CaCl2, and 10 mM MgSO4, pH adjusted to 7.3 with HCl, osmolarity 305 mOsm, and saturated with 95% O2/5% CO2). Coronal slices (250 μm) were cut with a vibratome (Leica VT1200s) and allowed to recover first in oxygenated NMDG-HEPES at 37 °C for 9 min, then transferred to HEPES-aCSF (86 mM NaCl, 2.5 mM KCl, 1.2 mM NaH2PO4, 35 mM NaHCO3, 20 mM HEPES, 2 mM CaCl2, 1 mM MgSO4, 25 mM glucose, 5 mM sodium ascorbate, 2 mM thiourea, and 3 mM sodium pyruvate, pH adjusted to 7.3, osmolarity 305 mOsm) at room temperature for at least one hr. Recovered slices were transferred to a recording chamber mounted on an Olympus BX61WI microscope, under continuous perfusion (2 ml/min) with oxygenated artificial CSF (aCSF: 119 mM NaCl, 2.5 mM KCl, 1 mM NaH2PO4, 1.3 mM MgCl2, 2.5 mM CaCl2, 26.2 mM NaHCO3, and 11 mM glucose, osmolarity 290 mOsm, saturated with 95% O2/5% CO2) heated to 29 ± 2 °C via an in-line heater (Warner Instruments). Whole-cell recordings were made from visually identified mPFC L2/3 pyramidal neurons. The extracellular solution contained TTX (1 μM, tetrodotoxin citrate, Hello Bio) and D-AP5 (100 μM, Hello Bio) to block action potentials and NMDA receptor activities respectively. The intracellular solution contained 108 mM Cs-gluconate, 20 mM HEPES, 5 mM TEA-chloride, 2.8 mM NaCl, 0.4 mM EGTA, 5 mM BAPTA, 4 mM MgATP, 0.3 mM NaGTP, pH 7.2 and 290 mOs. mEPSCs were recorded as inward currents at a holding potential of −72 mV, and mIPSCs were recorded as outward currents at a holding potential of 0 mV. After acquisition of a stable baseline (~8 min), miniature postsynaptic events were recorded for ~8 min at each potential. Recordings were excluded if the access resistance was >25 MΩ or changed >20% during the recordings. All signals were filtered at 2.6 KHz, amplified at 5x using a MultiClamp 700B amplifier (Molecular Devices), and digitized at 20 KHz with a Digidata 1440 A analog-to-digital converter (Molecular Devices). Patch-clamp data were acquired using Clampex 10.4 and analyzed using Clampfit 10.4 (pCLAMP 10.4 software suite, Molecular Devices).

**Table 1 | Genotype of pups at weaning from Pacs1$^{HET}$ x Pacs1$^{HET}$ breeding**

|   | Genotype | | | |
|---|---|---|---|---|
|   | Pacs1$^{WT}$ | Pacs1$^{HET}$ | Pacs1$^{KO}$ | Total |
| E | 74.75 | 149.5 | 74.75 | 299 |
| O | 78 | 168 | 53 | 299 |

E: Expected; O: Observed. $\chi^2 = 8.76$ $p = 0.013$.

**Table 2 | Genotype of pups at weaning from Pacs1$^{HET}$; Pacs2$^{HET}$ x Pacs1$^{HET}$; Pacs2$^{HET}$ breeding**

| N = 121 | Pacs1 alleles | | | | |
|---|---|---|---|---|---|
| Pacs2 alleles | | | Pacs1$^{WT}$ | Pacs1$^{HET}$ | Pacs1$^{KO}$ |
| | Pacs2$^{WT}$ | E | 7.56 | 15.13 | 7.56 |
| | | O | 13 | 18 | 7 |
| | Pacs2$^{HET}$ | E | 15.13 | 30.25 | 15.13 |
| | | O | 21 | 38 | 7 |
| | Pacs2$^{KO}$ | E | 7.56 | 15.13 | 7.56 |
| | | O | 5 | 12 | 0 |

E: Expected; O: Observed. $\chi^2 = 22.21$, $p = 0.0045$.

## Immunohistochemistry

Mice were anesthetized with ketamine (90 mg/kg)/ xylazine (20 mg/kg) and transcardially perfused with ice-cold 4% paraformaldehyde (PFA). Brains were removed, post-fixed overnight in 4% PFA at 4 °C, dehydrated in 30% sucrose, and embedded in OCT (Tissue Tek). 35 μm or 50 μm sections were prepared on a Leica CM1950 cryostat or a Leica VT1000S vibratome, respectively. Staining was performed on free-floating sections without antigen retrieval. Sections were washed 3x with 0.1% Triton-X100 in PBS and then blocked for 1 hr with 5% normal goat serum (NGS), 1% bovine serum albumin (BSA), and 0.3% Triton-X100 in PBS at RT. Primary antibodies were diluted in 0.1% Triton-X100 and 5% NGS, and sections were incubated overnight at 4 °C with gentle agitation. Sections were then washed 3x (10 min/wash) with 0.3% Triton-X100 and 1% NGS in PBS, and incubated with species-specific secondary antibodies diluted in PBS containing 1% bovine serum albumin (BSA), 2% NGS, and 0.1% Triton-X100 for 2 h at RT. Nuclei were counterstained with Hoechst 33342. Sections were mounted onto glass slides using Fluoromount-G (Invitrogen, 00-4958-02), cured, and imaged.

## Dendrite arbor reconstructions and spine density analysis

P1 pups were ICV-injected with 3 x 10$^8$ pAAV-FLEX-tdTomato viral particles (Addgene #28306-AAV9) combined with 40 μg of ASOs, as detailed above. At P18, mice were anesthetized with ketamine/ xylazine and transcardially perfused with ice-cold 4% paraformaldehyde (PFA). Brains were removed and post-fixed overnight in 4% PFA at 4 °C with gentle agitation. For dendritic reconstructions, 250 μm coronal slices were sectioned using a Leica VT1000S vibratome. Slices containing the hippocampus were incubated with a 1:800 Hoechst 33342 solution in PBS for 10 min at RT under gentle agitation and then washed 3X with PBS. Slices were subsequently mounted onto glass slides with Fluoromount-G. CA1 pyramidal tdTomato$^+$ neurons were identified based on the cell body location relative to other hippocampal areas. Full z-stack images were acquired on a Nikon A1R confocal Ti2-E microscope. Dendritic arbors were traced using the Simple Neurite Tracer (SNT) plugin of the Fiji software, as previously described[65]. Sholl analysis, total dendrite length, and branching point number were subsequently quantified using the same software. For spine density

analysis, 50 μm coronal slices were acquired on a vibratome. Sections containing the hippocampus were blocked for 1 hr in 5% normal goat serum (NGS), 1% BSA, and 0.3% Triton-X100 in PBS at RT. Sections were then incubated overnight at 4 °C with rabbit anti-RFP. Sections were washed three times for 10 min and incubated for 90 min with goat anti-rabbit Alexa Fluor 488. Brain sections were washed 3 times and counterstained with Hoechst 33342. Spines on secondary apical dendrites of CA1 hippocampal pyramidal neurons were imaged and analyzed with NeuroLucida360 and the dendritic length was calculated using NeuroExplorer 360 as described[66].

## Neuron Isolation and Culturing

Primary mouse hippocampal neurons were prepared from P0 $Emx1^{Cre/+}$; $R26^{P1}$ or $Emx1^{Cre/+}$; $R26^{P1R203W}$ mice as described[67] with a few modifications. Pups were anesthetized in ice for 9 min, brains were harvested, and hippocampi were micro-dissected. Excised hippocampi were washed and resuspended in ice-cold HBSS (Gibco, 14175-095) supplemented with 1 mM sodium pyruvate (Gibco, BW13115E), 0.1% w/v glucose (Sigma, G6152) and 10 mM HEPES pH 7.3 (Gibco, 15630-080), followed by a 15 min digestion by papain (Worthington, LK003178) and 6 μg/ml DNase I (Sigma, DN25) supplemented with 2 mM MgSO4 (Sigma, M2643) at 37 °C with gentle agitation. Dissociated cells were pelleted and resuspended in NeuroBasal medium (Gibco, 21103049), gently triturated using polished Pasteur pipettes, and filtered through a 70 μm nylon strainer. Cells were counted using Trypan Blue, washed, and resuspended in NeuroBasal maintenance medium supplemented with B-27 (Gibco, 17504044), GlutaMax (Gibco, 35050061), and penicillin/streptomycin (Lonza, 17-602E). 4 x 10$^4$ cells were seeded onto 18 mm coverslips washed with 70% nitric acid (J.T. Baker, #959800) and thinly coated with Matrigel (Corning, 354277) in a 24-well plate. Six hr after plating, half of the medium from each well was carefully removed and replaced with fresh maintenance medium. On DIV2 (2 days post-plating), cells were treated with 2 μM AraC (Sigma, C6645) to reduce glial cell growth. The maintenance medium was replenished every 3–4 days until cells were processed for imaging.

## Confocal microscopy

Images were captured using a Nikon Ti2-E confocal microscope with a resonant scanner and processed using the Nikon Elements analytical package. Golgi localization (juxtanuclear vs peripheral) and fragmentation analyses were designed and automated using Nikon Elements software. Deconvolved multi-channel images were segmented using automatic Otsu thresholding. An ROI dividing the cytosolic compartments was automatically generated in each cell with a perimeter set 10 μm (dermal fibroblasts), 4 μm (MEFs), 3 μm (Hela cells and NPCs) from the edge of the nuclear mask. The juxtanuclear/peripheral border was automatically determined and dependent on the shape and size of the nucleus of each cell. Golgi dispersion was determined by the position of Golgi objects (percentage of total Golgi area) relative to the ROI. Juxtanuclear Golgi included all Golgi objects located between the margin of the ROI and the nuclear envelope. Peripheral Golgi included all objects outside the ROI border. Colocalization of PACS1/PACS1$^{R203W}$ with HDAC6 was quantified through Pearson's coefficient (R) using the Coloc2 plugin integrated in Fiji (ImageJ). Deconvolved z-stack images were segmented to identify high-density bodies based on the intensity of the fluorescent signals, the object area (>2 μm$^2$), and their circularity. The bodies were then analyzed for the spatial correlation (colocalization) between PACS1/PACS1$^{R203W}$ and HDAC6.

## Golgi topology 3D analysis of primary neurons

High-resolution tiled images of entire cultured hippocampal neurons were captured using a Nikon Ti2-E confocal microscope equipped with high-sensitivity GaAsP detectors and with a 60x oil immersion objective and 3x zoom (pixel size (XY) = 0.069 μm/pixel, Z step = 0.15 μm).

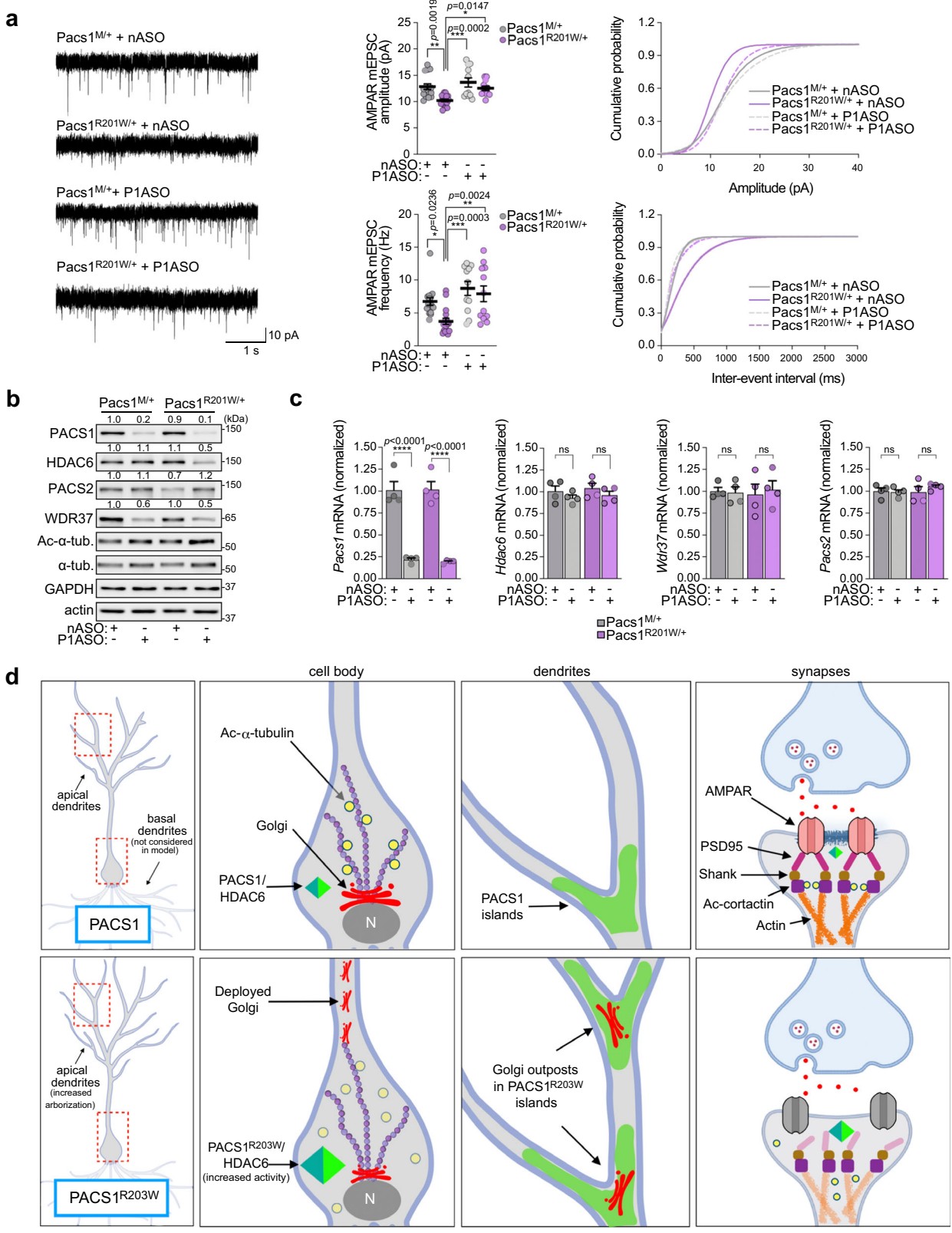

All images were deconvolved with Huygens Professional version 22.10 (Scientific Volume Imaging, The Netherlands), using the CMLE algorithm. Deconvolved images were saved in ICS format to ensure optimal preservation of the image metadata (32-bit). Three-dimensional (3D) analysis of the Golgi topology was performed using Imaris 10.0.0 software (Bitplane, Zurich, Switzerland). Golgi of interest was identified and masked as the positive voxels (red channel) contained within the cultured neurons expressing the HA-tagged PACS1 or PACS1^R203W proteins (green channel). To identify and classify discrete Golgi objects, the Surface creation tool was used. Surface classification was optimized by applying a constant background subtraction threshold across all images. Subsequently, Golgi surfaces were filtered to exclude very small disconnected objects ($<0.003\ \mu m^3$). Golgi surfaces positioned within a distance of $2.5\ \mu m$ from the nucleus were classified as Somatic Golgi whereas objects distal to that cutoff were classified as Neurite Golgi.

**Fig. 6 | PACS1 ASO rescues PACS1$^{R201W}$-induced deficits in excitatory synaptic communication. a** (Left) Representative whole-cell voltage-clamp recordings of mEPSCs from L2/3 pyramidal neurons in acute brain slices of juvenile *Emx1$^{Cre}$*-induced Pacs1$^{R201W/+}$ and Pacs1$^{M/+}$ control mice injected (ICV) at P1 with 40 µg nASO or P1ASO. (Middle) AMPAR mEPSC amplitude (top) and frequency (bottom). (Right) Cumulative probability distributions of AMPAR mEPSC amplitudes (top) and frequencies (bottom). Data are mean ± SEM (2-way ANOVA followed by Tukey *post hoc* test). *n* = 16 neurons/6 mice (Pacs1$^{M/+}$/nASO), 17 neurons/4 mice (Pacs1$^{R201W/+}$/nASO), 11 neurons/5 mice (Pacs1$^{M/+}$/P1ASO), 12 neurons/4 mice (Pacs1$^{R201W/+}$/P1ASO). **b** Western blots of cortical lysates from mice in panel a. Numerical values depict normalized mean signal intensity of the indicated bands, *n* = 4 mice in three independent experiments (graphs shown in Fig. S7d). **c** qRT-PCR of cortex RNAs encoding PACS1, PACS2, HDAC6 and WDR37 isolated from the ASO-treated *Emx1$^{Cre}$;Pacs1$^{M/+}$* and *Emx1$^{Cre}$;Pacs1$^{R201W/+}$* mice. Data are mean ± SEM (2-way ANOVA followed by Tukey *post hoc* test), *n* = 4 animals/condition. *p*-values shown in figure. Primers shown in Table S2. **d** Working model for the effect of PACS1$^{R203W}$ on neuronal structure and function. (Left) PACS1 syndrome neurons (PACS1$^{R203W}$) exhibit increased dendrite arborization. Red boxes denote cell bodies and dendritic branch points shown in the center panels. (Center left) In healthy (PACS1) neurons, acetylated (yellow circles) MTs stabilize the Golgi ribbon in the cell body. In PACS1 syndrome neurons, PACS1$^{R203W}$/HDAC6 causes excessive deacetylation of MTs, leading to Golgi fragmentation and dissemination into dendrites. (Center right) In PACS1 syndrome neurons, Golgi outposts frequently associate with PACS1$^{R203W}$ islands, which collect in dendritic varicosities and branch points. The disseminated Golgi outposts in PACS1 syndrome neurons are coupled to dendritic overbranching (left panels). (Right) In healthy neurons, Ac-cortactin stabilizes the PSD95-containing subsynaptic scaffold that organizes functional cell-surface AMPA receptors. In PACS1 syndrome neurons, PACS1$^{R203W}$/HDAC6 causes excessive deacetylation of cortactin, disorganizing the postsynaptic scaffold and reducing AMPA receptor-mediated mEPSCs. It is not known if PACS1$^{R203W}$/HDAC6 causes lateral diffusion or internalization of the affected AMPARs. Potential roles for PACS2 or WDR37 in these processes are not considered. Created with Biorender.com. Source data are presented as Sources Data files.

## qPCR

RNA was isolated from micro-dissected mouse cortex using RNeasy Mini Kit (QIAGEN, 74104). cDNA was prepared using SuperScript IV First-Strand Synthesis System Kit (Invitrogen, 18090050). qPCR reactions were performed in a QuantStudio 3 Real-Time PCR System (Applied Biosystems, A28567) with the PowerUp SYBR Green PCR Master Mix (Applied Biosystems, A25776) and the primer pairs listed in Table S1. All reactions were performed in triplicate. qPCR primers used in this study are listed in Tables S1 and S2.

## Immunoprecipitation and Western blot

For most immunoprecipitation experiments, HCT116 cells were transfected with plasmids using lipofectamine 2000. After 24 hr, cells were lysed in harvest buffer (50 mM Tris-Cl pH 7.4, 150 mM NaCl, 1% NP-40, and 10% glycerol) containing protease inhibitors (0.5 mM PMSF, 0.25 mM Pefabloc, 0.1 mM Aprotinin, 3 µM E-64 and 0.1 mM Leupeptin) and phosphatase inhibitors (1 mM Na$_3$VO$_4$ and 20 mM NaF). Lysates were precleared at 16,000 x g for 15 min, and 10% of the supernatant was mixed with 5x Laemmli sample buffer as input control, while the remaining lysate was incubated with a 50% slurry of anti-flag M2 affinity beads (Sigma, A2220) at 4 °C overnight. The beads were washed three times with modified RIPA buffer (50 mM Tris-Cl pH 8.0, 150 mM NaCl, 1% NP-40, and 1% DOC), and captured proteins were eluted in 2x Laemmli sample buffer at 95 °C for 5 min. The input control and the IP eluate were separated on SDS-PAGE gels for Western blot analyses. Western blots were developed with the Pierce ECL Western Blotting Substrate (Thermo Fisher, 34580) using a FluorChem E image acquisition system (ProteinSimple). Signals were quantified using the AlphaView image analysis software package (ProteinSimple).

## Brain harvest and lysate preparation

Animals were anesthetized with isoflurane and then decapitated. Brains were removed, immediately frozen in liquid nitrogen, and stored at −80 °C until further processing. Cortices were homogenized in ice-cold RIPA buffer (1% Triton-X100, 0.1% SDS, 1% DOC, 50 mM Tris-HCl pH 7.4, 150 mM NaCl and 1 mM EDTA) containing freshly added protease/phosphatase inhibitors. For acetylated protein blotting, 5 µM trichostatin A (TSA) was added to the lysis buffer. Protein concentration of the brain lysates was determined by the Bradford method according to the manufacturer's instructions (Bio-Rad 500-0205). For Western blots, 15-30 µg of protein was separated on 6-12% SDS-PAGE gels.

## Synaptosome fractionation

Synaptosome fractions were prepared essentially as described[68]. Whole brains were harvested from C57BL/6 mice and homogenized in 50 mM Tris-Acetate pH 7.4, 10% w/v sucrose, and 5 mM EDTA containing protease inhibitors (0.5 mM PMSF, 0.1 mM Aprotinin, 0.25 mM Pefabloc, 3 µM E-64 and 0.1 mM Leupeptin) and phosphatase inhibitors (1 mM Na$_3$VO$_4$ and 20 mM NaF) using a Teflon glass homogenizer. Lysates were centrifuged at 800 x g for 20 min, and the supernatant (cytosol fraction) was transferred to a new tube and sedimented at 16,000 x g for 30 min. The supernatant was saved as (non-synaptosomal fraction), and the pellet was resuspended in hypotonic buffer (5 mM Tris-Acetate, pH 8.1), incubated on ice for 45 min, homogenized in a Teflon glass homogenizer and mixed with 51% w/v sucrose to a final concentration of 34% w/v sucrose. An equal volume of 28.5% and 10% w/v sucrose cushion in 50 mM Tris-Acetate (pH 7.4) was sequentially layered onto the extract, and the 3-layer gradient was sedimented at 60,000 x g for 2 hr. The protein-enriched layer was collected at the interface between the 34% and 28.5% w/v sucrose layers, diluted with 50 mM Tris-Acetate pH 7.4 to 10% w/v sucrose. The sample was sedimented at 48,000 x g for 30 min, and the pellet was resuspended in 50 mM Tris-acetate pH 7.4 plus 0.1% SDS (synaptosomal fraction). Protein concentration was determined using the Bradford method according to the manufacturer's instructions (Bio-Rad, 500-0205). Four micrograms of total protein from each fraction were separated on 10% SDS-PAGE gels for Western blot analyses.

## Cell lines, antibodies, and chemicals

Cell lines- HCT116 and U2OS cells were maintained as described[59]. PACS1 syndrome dermal fibroblasts GM27159 (R203W patient), GM27160 (parent), GM27650 (R203W patient), and GM27651 (parent) were from Coriell Institute and immortalized by transduction with hTERT (LVP1130-Puro, Gentarget). CRISPR/Cas9 Hela PACS1$^{R203W}$ cells and the isogenic WT control line were generated by System Biosciences (CS721B-1). Pacs1$^{WT}$ and Pacs1$^{Δ4bp/Δ4bp}$ embryonic fibroblasts were isolated from E13.5 littermate embryos and immortalized with a retrovirus expressing SV40 large T antigen (kindly provided by M. Suda, UPMC). All cell lines were passaged in DMEM + 10%FBS and pen/strep. Antibodies- actin (Millipore, MAB1501, 1:3000), α-actinin (Cell Signaling Technology (CST) 3134 S, 1:1000), α-tubulin (DMA1 Cell Signaling 3873 S 1:1000 and Thermo Fisher 66031, 1:250), Ac-Lys$^{40}$-α-tubulin (CST 5335 S, 1:1000), cortactin (4F11 Sigma 05-180, 1:1000), Ac-cortactin (Sigma 09-881, 1:1000), CTIP2 (25B6 Abcam 18465, 1:500), SATB2 (Abcam 51502, 1:50), EB1 (BD Biosciences 610534, 1:50), V5 (Invitrogen, R960-25, 1:2000), HDAC6 (Abcam 253033, 1:50, D2E5 CST 7558 S 1:1000, and Assay biotech C0226, 1:1000), p62 (Abcam 56416, 1:100), Flag (Sigma-Aldrich, F7425, 1:5000 and A2220, 50% slurry), HA (CST 3724 S, 1:4000 and Biolegend 901513, 1:1000), furin (MON-152, kindly provided by J. Creemers, Leuven, 1:100), GAPDH (14C10, CST 2118 S, 1:1000), Giantin (kindly provided by Dr. A. Linstedt, CMU, 1:750), GM130 (BD Biosciences 610534, 1:500), pericentrin (AbCam

4888, 1:500), MAP2 (Biolegend 801810, 1:5000), βIII-tubulin (Biolegend 801213, 1:500), Nestin (AbClonal A11861, 1:100), Pax6 (CST 60433, 1:200), Sox2 (D6D9, CST 3579, 1:400), PSD95 (NeuroMab 75-028-020, 1:250), GABA$_A$Rα1 (NeuroMab 75-136-020, 1:1000), AMPAR1 (CST 13185, 1:1000), WDR37 (Sigma HPA037565, 1:1000), RFP (Rockland 600-401-379, 1:800), PACS1 (BD Biosciences 611371, 1:100, Invitrogen PA558589, 1:100 and Ab 703[69], 1:1000), PACS2 (Ab 193[59], 1:1000), Goat anti-Rabbit IgG Alexa Fluor 488 (Invitrogen A11008, 1:400), Goat anti-Mouse IgG1 Alexa Fluor 568 (Invitrogen A11004, 1:400) Goat anti-Mouse IgG Alexa Fluor 647 (Invitrogen A-21242, 1:400), Goat anti-Chicken IgY Alexa Fluor 633 (Invitrogen A-21103, 1:400), Goat anti-Rat IgG Alexa Fluor 488 (Invitrogen A-11006, 1:400), Goat anti-Rabbit IgG Alexa Fluor 594 (Invitrogen A-11012, 1:400). Chemicals- Nocodazole (Sigma 487929), Tubacin (Cayman Chemical NCO778559), SW-100 (MCE HY-115475), ACY-1215 (MCE HY-16026), AGK2 (Sigma A8251), TTX (Hello Bio HB1035), D-AP5 (Hello Bio HB0225; Tocris Cat. No. 0106).

### Patient-derived iPSCs and NPCs

Patient 159 (PACS1 1002i-GM27159) and parent 160 (PACS1 1001i-GM27160) fibroblast-derived iPSCs were from WiCell. Four independent clones of 159 and 160 iPSCs were grown on Matrigel (Corning 354277)-coated plates in mTeSR™ medium (STEMCELL 05825). To generate NPCs, dissociated iPSCs cells were plated on poly-L-ornithine (PLO)/laminin-coated plates and grown as monolayers in STEMdiff™ Neural induction medium (STEMCELL 05833). Differentiation into NPCs was confirmed by staining for Nestin, Pax6, and Sox2.

### Plasmids, siRNAs, and AAV

Plasmids- pHDAC6 flag was kindly provided by J. Hu (University of Pittsburgh), pEGFP-Tubulin K$^{40}$Q and K$^{40}$R were gifts from Kenneth Yamada (Addgene plasmids #105302 and #105303; RRIDs: Addgene_105302 and Addgene_105303), pSIRT2 Flag was a gift from Eric Verdin (Addgene plasmid # 13813; RRID: Addgene_13813), pPACS-1-FLAG and pPACS-1-HA have been described[69]. siRNAs- Human PACS1 siRNA (L-006697-01), human HDAC6 siRNA (L-003499-00) and the non-targeting control RNA (D-001810-10, (Horizon/Dharmacon ON-TARGETplus)) were nucleofected (Amaxa) into cells as described[59]. AAV- pAAV-FLEX-tdTomato was a gift from Edward Boyden (Addgene viral prep # 28306-AAV9; RRID: Addgene 28306). Flag-tagged PACS1$^{R203W}$ was made by site-directed mutagenesis using standard PCR methods. Other plasmids were generated by Gibson Assembly (GA) cloning using GA reaction mix (NEB M5510AA) and gBlocks (IDT). These include HDAC6-V5 as well as FLAG-tagged HDAC6$^{1-465}$, HDAC6$^{1-864}$, HDAC6$^{Δ4-85}$, HDAC6$^{Δ4-476}$ (see Fig. 1 panel I, constructs B-E, respectively), and also Flag-tagged PACS1$^{Δ5-542}$, PACS1$^{1-542}$, PACS1$^{1-266}$, PACS1$^{Δ5-117}$, PACS1$^{1-117}$ and PACS1$^{1-266R203W}$ (see Fig. 1j constructs B-F and Fig. S2c, respectively).

### HDAC6 activity

HDAC6 activity was measured essentially as described[70]. Briefly, HDAC6/f was immunoisolated from transfected cell lysates using anti-FLAG M2-affinity gel (Millipore A2220). The beads were washed and bound HDAC6/f was eluted with FLAG peptide (Millipore F3290). Following normalization of immunoreactive protein by Western blot, HDAC6/f-containing eluants were incubated with 50 μM FLUOR-DE-LYS peptidyl substrate (Enzo BMLKI1040050) for 2 hr at 37 °C in activity buffer (140 mM HEPES, 10 mM KCl, 1 mg/ml BSA 1 mM βME, pH 7.4). Fluorescent AMC was released from the deacetylated substrate by digesting with Trypsin-TPCK (Sigma T1426) in resolving buffer (20 mM Tris, 150 mM NaCl, 1 mM EDTA, pH 7.4). AMC signals were detected using a BioTek Synergy 4 microplate reader ($\lambda_{ex}$ = 420 nm) and ($\lambda_{em}$ = 460 nm). Controls included no HDAC6/f or addition of 5 μM tubacin.

### Statistics and reproducibility

Graphpad PRISM was used to calculate the statistical significance of differences and the statistical test used for each experiment is described in the respective figure legends.

### Availability of materials

Patient cell lines are available from Coriell Institute. Patient iPSCs are available from WiCell. Hela CRISPR/Cas9 PACS1$^{R203W}$ knock-in cells and their isogenic control are from System Biosciences. ASOs are from IONIS. All plasmids, ASOs, and mouse lines developed during this study will be shared with proper material transfer agreements in place with the University of Pittsburgh.

### Reporting summary

Further information on research design is available in the Nature Portfolio Reporting Summary linked to this article.

## Data availability

All data generated or analyzed during this study are included in the published article and its supplementary file. Source data are provided with this paper and with Figshare https://figshare.com/s/a3f336dca03225fe2558. Source data are provided with this paper.

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

## Acknowledgements

This work was supported by NIH grants DK112844, DK114855, and NS123649, NORD, PACS1 Syndrome Research Foundation (to GT) as well as a University of Pittsburgh Health Sciences Diversity Scholars Fellowship (to JBL). We are particularly grateful to the PACS1 syndrome patients and their families for their generous donation of tissues and cells. We thank Drs. S. Rizzo, R. Peixoto, L. D'Aiuto, B. Molyneaux, U. Pandey, and S. Kour for helpful discussions, S. Huygens and T. Gebler for logistics, T. Brosenitsch for editing of the manuscript, and J. Cantine for video editing. We thank Drs. A. Linstedt (CMU), J. Creemers (Leuven), and M. Suda (Pitt) for reagents, and Dr. Lee (USCD) for subcloning into the pR26-GFP-Dest vector. Special thanks to Dr. C. Bi and Z. Kou of the Mouse Embryo Services Core for microinjection of zygotes and production of the new mice strains, and also to A. Marquez, B. Meola, M. Castello, R. Kielty, and M. Maanas for assistance with the mouse colony.

## Author contributions

S.V.-P., L.T., Y.Y., Y.H.H., and G.T. designed the study. S.V-P., L.T., Y.Y., K.C., J.B.L., B.D., and M.J.G. conducted the experiments. S.G. designed and developed the mouse lines. J.O., K.L., B.P., and H.B.K. screened and validated the ASOs. G.T. prepared the manuscript with editing by Y.H.H., Y.Y., S.V-P., S.G., and M.J.G. L.T. designed and prepared the artwork.

## Competing interests

BP, HBK, JO, and KL are employees and shareholders of Ionis Pharmaceuticals. The other authors declare no competing interests.
