## [Peer Review File · Nature Communications]

Neural deficits in PACS1 syndrome caused by dysregulated HDAC6 are corrected with targeted therapyReviewers' Comments:

Reviewer #1:

Remarks to the Author:

In this manuscript, Villar-Pazos and colleagues used a combination of molecular and cellular analyses on patient-derived cells in vitro and transgenic mice to study the effects of a missense mutation in the PACS1 gene, which encodes a multi-functional trafficking protein and is linked to a neurodevelopmental genetic disorder characterized by intellectual disability.

First, the authors performed immunostaining on fibroblasts derived from one patient bearing the R203W PACS1 mutation to show that the Golgi apparatus is disorganized in these patient cells when compared to the respective parental control. They also conducted microscopy and biochemistry assays (Western Blotting and co-IP) to show that the microtubules are disorganized in the patient fibroblasts, an effect which the authors concluded to be the result of tubulin deacetylation led by increased HDAC6 activity. The authors then switch models and perform a series of experiments to assess PACS1 function and cellular abnormalities in the neural tissue of genetically modified mice. First, they achieved expression of a R203W PACS1 transgene from the ROSA26 locus and used immunostaining to show that the mutant protein is mis-localized at the base of neurites and inside growing dendrites and axons. They also show that the dendritic arborization pattern is exacerbated in mouse hippocampal cells expressing the mutant protein, an aberrant phenotype accompanied by altered electrophysiological parameters in mouse brain slices, both of which can be corrected via ASO-mediated knock-down of HDAC6 expression. In another line of experiments, the authors created a knock-in mouse line in which the PACS1 mutation found in patients is conditionally introduced in brain cells. In these animals, the loss of electrophysiological parameters is similar to that observed in the ROSA mice, a condition that can be reversed by ASO-mediated knock-down of PACS1 expression.

This is an interesting study, which attempted to decipher cellular abnormalities linked to a specific disease-associated mutation, with a therapeutic twist by the inclusion of proof-of-concept experiments in which the expression of one or more target genes is manipulated via ASO treatment. Although the authors explore the consequences of R203W PACS1 mutations on several potentially relevant cellular phenotypes, it remains to be determined if the causality relationships proposed here indeed apply to the human neural tissue, as some pieces of the disease mechanistic model presented in Fig. 4 were obtained by investigating fibroblast cells (patient- or mouse-derived), not neural cells, as detailed below.

Major comments:

1. My main criticism is that the narrative constructed around a Golgi defect that is accompanied by microtubule disorganization, which in turn leads to aberrant synapse formation and function, is drawn from bits and pieces derived from different types of cells/models. Some experiments are conducted with patient-derived fibroblasts (which, though irrelevant for understanding the neural phenotypes, may be a good starting point for creating testable hypotheses for the neuronal defects) while others are conducted with mouse neural tissue, but the manuscript fails to convincingly demonstrate which altered phenotypes in the model presented in Figure 4 are exhibited by the neural tissue and why. One example of this is the Golgi distribution aberrations observed in fibroblasts. When the authors try to determine whether these alterations would also be present in neural cells in vitro (from the mouse models), the results are not convincing. Figures S3c and 2d attempt to show Golgi mis-localization, but the immunostaining images are low-mag and I am not convinced that the diffuse pattern of Golgi marker localization in these experiments can be interpreted as fragmented or mis-localized Golgi, as the authors claim. Moreover, I do not see much difference between the R26-P1 and R26-P1R203W panels in Fig. S3c, even though the quantifications on the right make it appear as though the differences are clear cut. Better quantification methods for assessing Golgi mis-localization must be used, such as high-resolution confocal microscopy followed by 3D reconstruction or subcellular fractionation followed by biochemical sedimentation analysis.
2. Importantly, the links between Golgi mis-localization and the defects in neuronal morphology

and/or physiology are not properly established. If the model is that the Golgi impairment leads to aberrant neurite growth (as Fig. 4 seems to suggest), then a better assessment of Golgi localization must be performed, and causality experiments that correct both Golgi localization and neuronal function must be performed in the neural tissue, either from the mouse models and/or from patient-derived neurons in vitro (obtained from patient iPSC lines, for example).

3. Another important criticism is that the study tries to model PACS1 syndrome by using cultured cells from a limited number of patient donors. Most experiments are conducted with a single patient/parental control pair, and a second pair is presented as additional evidence (not as biological replicate) in a few experiments. This is much less than ideal and probably insufficient. Given the notorious variability across cell lines from human subjects (patient or controls), this approach must be complemented by the addition of new cell line pairs and/or pairs of isogenic lines (patient cells with corrected mutation and/or parental control cells with the R203W mutation, plus respective controls). I acknowledge that this requires additional work, but it is needed in this case if any meaningful and generalizable conclusions are to be drawn from these human cellular models.

4. The authors try to link the R203W PACS1 mutation to HDAC6 and tubulin defects. In another part of the manuscript, the mechanistic model includes WDR37 and PACS2 (whose mechanistic participation in PACS1 function was suggested by another publication), but the paper fails to provide a model in which both observations are reconciled. If a subset of the cellular phenotypes is the result of defects in one pathway while other alterations are due to changes in other mechanistic components, then this must be shown (preferably in neural cells) and properly documented, otherwise the model proposed in Fig. 4g is not appropriate to explain the consequential neuropathology.

Minor comment:

- The title is misleading, as the authors have done more than just perform RNA-directed therapy and more than just study a mouse model for PACS1 syndrome.
- In figure 2c, some type of quantification of HA signal on R26 P1R203W apical dendrites should be presented, since there appears to be no substantial difference between the two images.

Reviewer #2:

Remarks to the Author:

This paper describes an RNAi based method to restore neuronal deficits in a PACS1 syndrome mouse model. Overall, the data presented is of high quality, but several important conclusions require additional experiments. The lack of an introduction and highly truncated discussion made this manuscript difficult to interpret. Major and minor concerns are listed below:

MAJOR

1) Significance for this work is drawn from the potential for novel therapeutics for PACS1 syndrome. However, no data shows that the mouse models created represent a reasonable model of the disease. Constitutive knockin of the mutation is lethal, which does not recapitulate the human condition. Given the poor translatability of animal research, a reasonable attempt should be made to show that the transgenic mice created model the disease, at least to some extent. This could be behavioral assays or measurements of neuronal activity in patient neurons, which could be obtained from direct-conversion of fibroblasts.

2) Additional experiments are needed to bolster the major and central conclusion, which is that increased interactions between mutant PACS1 and HDAC6 drive PACS syndrome. The only evidence presented is co-immunoprecipitations from heterologous cells that overexpress proteins, which is insufficient. Patient cells present the perfect opportunity to test the hypothesis that R203W increases the association of PACS1 with HDAC6. It is surprising that coimmunoprecipitation and colocalization of endogenous proteins were not done since antibodies for endogenous proteins are available.

3) Given the focus on neuronal activity and known differences in golgi organization between cell types, mechanistic studies should also be done in neurons. While patient fibroblasts could be converted to neurons, an isogenic control should always be included to control for the differences in genetic

architectures, even among family members.

4) Measurements of Golgi localization should be controlled for the size of the cell and polarity of the golgi stack. Patient cells seem to be larger (Fig1,S1), so one could expect that their perinuclear distance is also affected given the absolute measurements described in the methods. Given the proposed disruptions to microtubule acetylation, one would expect changes in cellular morphology.

5) The electrophysiology results are difficult to reconcile with the model. The authors argue that ultimately, HDAC6-dependent acetylation alters pre and postsynaptic function. However, HDAC6 knockdown does not alter neuronal morphology (Fig 2F) or synaptic activity (Fig 3a). Moreover, the PACS1 KO mice show elevated Acetyl-tubulin (Fig S5d), but do not show synaptic deficits. These results undermine the central hypothesis.

MINOR/SPECIFICS

1) At least one other golgi marker should be used to show that PACS1 is not altering the localization of Giantin, which is typically localized to cis-Golgi.

2) Authors should determine and quantify whether wt or mutant PACS1 alter HDAC6 overall expression. Some figures seem to suggest this (Fig1F), which would significantly change the interpretation of results (Fig 1J).

3) Authors state that there is more PACS1R203W in dendrites (Fig 2c). This reviewer doesn't see this, so it should be quantified.

4) In general, immunocytochemistry of neuronal cultures is poor (Fig 2d). Golgi structures should be clearly visible and do not normally fill the cytoplasm as shown. PACS1 also does not seem to localize to Golgi as expected. Authors report golgi fragments in dendrites, but these are almost certainly axons. This would be odd as neuronal axons should not have golgi structures.

5) Reconstructions of neuronal morphology from Tdtomato-filled AAV-infect neurons is impressive. Images should be included as this reviewer does not understand how this could be done without sparse labeling of neurons (typically done with a different virus).

6) Synaptosomal fractions are notoriously contaminated. The protein expression patterns shown in Fig S4a (decreasing amount) are typically interpreted as being absent from synapses.

7) 5-10 hz frequencies for Minis is very high. An explanation should be provided for these results.

8) The liquid phase separation data is superficial and does not contribute to the main hypothesis.

Reviewer #3:

Remarks to the Author:

This is a comprehensive translational study that elucidates further the cellular defects underlying PACS1 syndrome; reports new mouse models of the recurrent PACS1-R203W causal variant; and tests pharmacological and ASO-based therapies on neuronal structure and synaptic transmission in vivo. Using patient and parent fibroblasts, the authors show that PACS1-R203W disrupts Golgi positioning and microtubule organization, offers functional links to α -tubulin acetylation and show that R203W increases the interaction with HDAC6. Pharmacological and genetic HDAC6 inhibition were able to rescue these cellular defects significantly, prompting validation in vivo. Using a newly generated mouse model of PACS1 syndrome with a humanized conditional allele bearing PACS1-R203W in the Rosa26 safe harbor locus, the authors expressed WT or variant PACS1 in Emx1-Cre-driven neurons. The Golgi positioning and dendrite morphology, as well as synaptic function, were also altered in PACS1-R203W mouse models, defects that could be rescued significantly with HDAC6 ASOs. Finally, the authors pursued the possibility of targeting PACS1-R203W directly with ASOs through generation of Pacs1 KO and conditional Pacs1-R201W KI mouse models. Overall, the manuscript is original, represents a substantial amount of work to advance our knowledge of the molecular basis of PACS1 syndrome, highlights the PACS1-HDAC6 interaction as a key contributor and to pathology, and excitingly, shows that the syndrome may be treatable with either HDAC inhibitors or ASOs. The methodology is well described and the data are beautifully presented in each figure.

I am enthusiastic about this work advancing toward publication and have only a few comments.

1. Please show dot plots on all charts, or at least overlay them onto the bars. This is particularly important for the n=3 datasets.
2. Figure 2c shows differential subcellular localization of HA-tagged PACS1 and PACS1-R203W. First, it is really difficult to see this difference given the bright signal throughout the section so please modify the presentation of the image to enable the reader to more readily see this difference. Additionally, please show how the ha-tag of the transgene compares to localization of endogenous WT PACS1 with a co-stain.
3. Please quantify the protein levels in Figure 4b to better characterize the impact on HDAC6 and WDR37.
4. The notable rescue of Golgi positioning with not one, but three different HDAC inhibitors as shown in Figure S2 is underplayed in the main text and should be emphasized further.
5. Although this is already an extensive dataset, one piece that would really complement this story is to know whether PACS1-R203 or Pacs1-R201 mouse models have been characterized through behavioral testing. While the structural and functional phenotype readouts are rational measures for therapeutic amelioration with HDAC6 inhibitors or ASOs, this would offer robust in vivo data to support further the authors' conclusions.

REVIEWER COMMENTS

Reviewer #1 (Remarks to the Author):

In this manuscript, Villar-Pazos and colleagues used a combination of molecular and cellular analyses on patient-derived cells in vitro and transgenic mice to study the effects of a missense mutation in the PACS1 gene, which encodes a multi-functional trafficking protein and is linked to a neurodevelopmental genetic disorder characterized by intellectual disability.

First, the authors performed immunostaining on fibroblasts derived from one patient bearing the R203W PACS1 mutation to show that the Golgi apparatus is disorganized in these patient cells when compared to the respective parental control. They also conducted microscopy and biochemistry assays (Western Blotting and co-IP) to show that the microtubules are disorganized in the patient fibroblasts, an effect which the authors concluded to be the result of tubulin deacetylation led by increased HDAC6 activity. The authors then switch models and perform a series of experiments to assess PACS1 function and cellular abnormalities in the neural tissue of genetically modified mice. First, they achieved expression of a R203W PACS1 transgene from the ROSA26 locus and used immunostaining to show that the mutant protein is mis-localized at the base of neurites and inside growing dendrites and axons. They also show that the dendritic arborization pattern is exacerbated in mouse hippocampal cells expressing the mutant protein, an aberrant phenotype accompanied by altered electrophysiological parameters in mouse brain slices, both of which can be corrected via ASO-mediated knock-down of HDAC6 expression. In another line of experiments, the authors created a knock-in mouse line in which the PACS1 mutation found in patients is conditionally introduced in brain cells. In these animals, the loss of electrophysiological parameters is similar to that observed in the ROSA mice, a condition that can be reversed by ASO-mediated knock-down of PACS1 expression.

This is an interesting study, which attempted to decipher cellular abnormalities linked to a specific disease-associated mutation, with a therapeutic twist by the inclusion of proof-of-concept experiments in which the expression of one or more target genes is manipulated via ASO treatment. Although the authors explore the consequences of R203W PACS1 mutations on several potentially relevant cellular phenotypes, it remains to be determined if the causality relationships proposed here indeed apply to the human neural tissue, as some pieces of the disease mechanistic model presented in Fig. 4 were obtained by investigating fibroblast cells (patient- or mouse-derived), not neural cells, as detailed below.

RESPONSE to introductory comment: We thank the reviewer for these comments and for the opportunity to present new data, including analysis of isogenic cell lines and patient-derived NPCs (see Figs. 1 and 4), which, together with an expanded introduction and discussion, indeed support the working model in Figure 6. Our study identifies PACS1 as a novel HDAC6 effector (Figs. 1, 2, S1, and S6), and that the R203W substitution represents a gain-of-function (GOF) mutation that increases the interaction between PACS1 and HDAC6 to aberrantly potentiate enzyme activity (Fig. 2g). Consequently, PACS1^{R203W}/HDAC6 causes neuronal deficits in PACS1 syndrome mice and in human neural cells. We show that PACS1 localizes to sites of HDAC6 action, including microtubules and dendritic spines (Figs. S1e and 5c), and that PACS1^{R203W}/HDAC6 reduces acetylation of α -tubulin and cortactin (Figs. 1 and 5), which play vital roles in dendrite arborization, spine density and synaptic communication. Consequently, the increased HDAC6 activity disrupts Golgi positioning in patient-derived NPCs and in the hippocampus, which is coupled with increased dendrite complexity (Figs. 3 and 4). The dendritic arbors, however, are beset with varicosities, diminished spine density and fewer functional synapses (Figs. 5 and 6), frequently found in other brain disorders, including IDD, autism and epilepsy, which we discuss. We

demonstrate the causality of PACS1^{R203W}/HDAC6 to these deficits both in patient-derived fibroblasts and NPCs, as well as in PACS1 syndrome mice, by application of highly selective HDAC6 inhibitors, as well as siRNA- or ASO-knockdown of PACS1 and HDAC6, which reverse the deficits in each of the model systems (Figs. 2, 4-6, and S2). As HDAC6 is a core component of both the membrane traffic and proteostasis control machineries in all tissues, we believe that interrogation of the PACS1 syndrome fibroblasts was critical to this study as it both informed and accelerated our discovery of a key role for PACS1^{R203W}/HDAC6 underlying neuronal deficits associated with PACS1 syndrome. In addition, the robust co-localization of PACS1/PACS1^{R203W} and HDAC6 in p62⁺ bodies only in patient cells (Figure 2d), suggests a broader impact of PACS1^{R203W} on HDAC6, including proteostasis control. The prevalence of dendritic varicosities in the PACS1 syndrome neurons supports this possibility (Figure 5a), which we also discuss.

In our revised Discussion, we relate our studies to the works of others to explain how the reduced acetylation of α -tubulin and cortactin lead to each of the neuronal deficits, and how in vivo HDAC6 modulators, including CAMDI, Septin7 and PACS1 may “tag team” to modulate sequential steps in cortical layering, neuronal arborization and synaptic communication. We further relate our electrophysiology work to published studies that also reported reduced acetylation of α -tubulin results in reduced mEPSC frequency whereas reduced cortactin acetylation reduces both spine density and mEPSC amplitudes (References 31 and 49).

Finally, while our work identifies PACS1^{R203W}/HDAC6 as a key driver of the neuronal deficits in PACS1 syndrome, we do not know if the fragmented Golgi is simply a consequent passenger or possibly as a second-level driver of the disorder. For example, the dispersed Golgi elements may function as secondary MTOCs that further increase dendrite complexity, or possibly as a rogue trafficking center that disturbs protein flux to distal sites. Testing these ideas requires uncoupling Golgi fragmentation from hypoacetylated microtubules, which is beyond the scope of this work.

Major comments:

1. My main criticism is that the narrative constructed around a Golgi defect that is accompanied by microtubule disorganization, which in turn leads to aberrant synapse formation and function, is drawn from bits and pieces derived from different types of cells/models. Some experiments are conducted with patient-derived fibroblasts (which, though irrelevant for understanding the neural phenotypes, may be a good starting point for creating testable hypotheses for the neuronal defects) while others are conducted with mouse neural tissue, but the manuscript fails to convincingly demonstrate which altered phenotypes in the model presented in Figure 4 are exhibited by the neural tissue and why. One example of this is the Golgi distribution aberrations observed in fibroblasts. When the authors try to determine whether these alterations would also be present in neural cells in vitro (from the mouse models), the results are not convincing. Figures S3c and 2d attempt to show Golgi mis-localization, but the immunostaining images are low-mag and I am not convinced that the diffuse pattern of Golgi marker localization in these experiments can be interpreted as fragmented or mis-localized Golgi, as the authors claim. Moreover, I do not see much difference between the R26-P1 and R26-P1R203W panels in Fig. S3c, even though the quantifications on the right make it appear as though the differences are clear cut. Better quantification methods for assessing Golgi mis-localization must be used, such as high-resolution confocal microscopy followed by 3D reconstruction or subcellular fractionation followed by biochemical sedimentation analysis.

RESPONSE to comment 1: We present in a new Figure 3d, application of Imaris analysis to the confocal images that were shown in the original Figure 2d (which have been moved to Figure S3C in the revised manuscript). These data clearly show a pronounced difference in Golgi positioning between

hippocampal neurons isolated from R26^{P1} versus R26^{P1R203W} mice. As expected, in R26^{P1} neurons, the Golgi remains localized to the cell body, with only a few Golgi elements found along a single neurite. By contrast, in R26^{P1R203W} neurons the Golgi was markedly fragmented and prominently deployed along every neurite. A quantification of these data is also presented in the new Figure 3d. We also note that PACS1^{R203W} is more extensively dispersed along each neurite compared to PACS1. We have included a .AVI video (Supplemental movie 1) for the reader to view the profound effect of the R203W disease substitution on the distribution of PACS1 and Golgi throughout the neurons. Note that the original figure S3c has been deleted.

2. Importantly, the links between Golgi mis-localization and the defects in neuronal morphology and/or physiology are not properly established. If the model is that the Golgi impairment leads to aberrant neurite growth (as Fig. 4 seems to suggest), then a better assessment of Golgi localization must be performed, and causality experiments that correct both Golgi localization and neuronal function must be performed in the neural tissue, either from the mouse models and/or from patient-derived neurons in vitro (obtained from patient iPSC lines, for example).

RESPONSE to comment 2: In a revised Figure 4c (adapted from the original Figure 2e), we prepared 50-micron slices of the specimens prepared from the TdTomato⁺/ASO-treated animals shown in the same panel to reconstruct dendritic arbors. Note that this arbor reconstruction required use of the Cre-inducible pAAV-FLEX-tdTomato at low m.o.i. to limit TdTomato expression of to only a subset of the (Cre-inducible) R26^{P1} or R26^{P1R203W} neurons. We stained the sections with anti-GM130 to identify Golgi and monitored TdTomato to identify and space fill the PACS1- or PACS1^{R203W}-expressing neurons. Consistent with our studies on the dissociated hippocampal neurons (Fig. 3d), the PACS1^{R203W}/nASO neurons harbored Golgi elements that extended into the TdTomato⁺ apical dendrite. By contrast, in the PACS1^{R203W}/H6ASO neurons, the Golgi returned to the cell body. As expected, neither PACS1/nASO nor PACS1/H6ASO affected localization of the Golgi to the cell body.

In a new Figure 4d, we show that PACS1^{R203W} disturbs Golgi positioning in patient-derived NPCs, similar to what we observed in R26^{P1R203W} hippocampal neurons (Figs. 3d and 4b). Treatment of the NPCs with tubacin for 24 hr rescued the Golgi to the juxtannuclear region. For this experiment, we obtained patient (159) and parent (160) iPSCs from Wi-Cell and differentiated them in NPCs using the monolayer protocol (STEMCELL). We have repeated this result in four different iPSC subclones. Invariably, on day 18, the patient NPCs spontaneously formed neurite-like processes containing deployed Golgi. In the revised Discussion, we state that similar findings have been observed in patient-derived iPSCs for other neurologic disorders, and which are associated with aberrant neurite outgrowth and dysregulated HDAC6 (refs. 42 and 43). Regardless, this experiment confirms that the neuronal deficits caused by PACS1^{R203W}/HDAC6 in the mouse models recapitulate the human disorder.

3. Another important criticism is that the study tries to model PACS1 syndrome by using cultured cells from a limited number of patient donors. Most experiments are conducted with a single patient/parental control pair, and a second pair is presented as additional evidence (not as biological replicate) in a few experiments. This is much less than ideal and probably insufficient. Given the notorious variability across cell lines from human subjects (patient or controls), this approach must be complemented by the addition of new cell line pairs and/or pairs of isogenic lines (patient cells with corrected mutation and/or parental control cells with the R203W mutation, plus respective controls). I acknowledge that this requires additional work, but it is needed in this case if any meaningful and generalizable conclusions are to be drawn from these human cellular models.

RESPONSE to comment 3: We obtained CRISPR/Cas9 knock-in heterozygous PACS1^{R203W} HeLa cells (Hela^{+P1R203W}) and their isogenic control from System Biosciences. We show in a new Figure 1b that PACS1^{R203W} causes the Golgi to fragment and disperse away from the MTOC region. As expected, in the isogenic control HeLa cells, the Golgi concentrates in the paranuclear region. This result supports our findings in the patient cell lines (Figs. 1a and S1a), and demonstrate that PACS1^{R203W} is causal to the disturbed Golgi positioning.

Our patient/parental HDF confocal study was limited to two cell pairs, which represented the entire inventory available from Coriell Institute. Note that we observed a similar result in fibroblasts from a third patient (161), but we lacked a parental control and so it was not included in our study. Anecdotally, we were struck by a statement from the PACS1 foundation regarding these data—the more severe the disease, the more dispersed the Golgi.

4. The authors try to link the R203W PACS1 mutation to HDAC6 and tubulin defects. In another part of the manuscript, the mechanistic model includes WDR37 and PACS2 (whose mechanistic participation in PACS1 function was suggested by another publication), but the paper fails to provide a model in which both observations are reconciled. If a subset of the cellular phenotypes is the result of defects in one pathway while other alterations are due to changes in other mechanistic components, then this must be shown (preferably in neural cells) and properly documented, otherwise the model proposed in Fig. 4g is not appropriate to explain the consequential neuropathology.

RESPONSE to comment 4: We agree that our handling of the PACS1/WDR37 work was awkward. The 2021 Nair-Gill et al paper (ref. 35) reporting the effect of PACS1/WDR37 on ER calcium dynamics was published during the course of our studies and given the potential role of ER calcium mishandling on mEPSCs, we could not ignore it. Note that the Nair-Gill paper stated that the PACS1/WDR37 deletion causes lymphopenia, which they explain has not been reported for patients afflicted with either PACS1 syndrome or WDR37 syndrome. Thus, there are no opposing observations from the different groups. We have recast the results and discussion to better explain these findings. In the revised Discussion on pages 12 and 13, we state:

“...Interestingly, PACS1 and WDR37 modulate calcium signaling by the ER, which is the cellular store that supplies calcium for action potential-independent mEPSCs^{35,53}. However, an aberrant PACS1^{R203W}/WDR37 interaction alone is unlikely to be the underlying cause of PACS1 syndrome deficits we report since the reduced mEPSC phenotype observed in R26^{P1R203W} mice was rescued by HDAC6 ASO, despite the presence of PACS1^{R203W} and WDR37 (Figs. 4b and see 5e).....”

Minor comment:

- The title is misleading, as the authors have done more than just perform RNA-directed therapy and more than just study a mouse model for PACS1 syndrome.

RESPONSE to minor comment 2: We have changed the title to “Neuronal deficits in PACS1 syndrome mice caused by PACS1^{R203W}-HDAC6 are corrected with RNA-targeted therapy”.

- In figure 2c, some type of quantification of HA signal on R26 P1R203W apical dendrites should be presented, since there appears to be no substantial difference between the two images.

RESPONSE to minor comment 2: We apologize for the unclear explanation of the immunohistochemical staining of HA-tagged PACS1 and PACS1^{R203W} in the CA1 region shown in Figure 3c. In the revised manuscript, we have labeled the images to mark the *stratum oriens* (OR), the *stratum pyramidale* (SP), representing the pyramidal cell layer, and *stratum radiatum* (SR), enriched in apical dendrites. We observed that PACS1 is much more concentrated in SP than is PACS1^{R203W}. Given that the two proteins are expressed at similar levels (see Fig. 3a), we speculated that PACS1^{R203W} may distend into the somatodendritic region or axon. The Imaris analysis of the dissociated hippocampal neuron in the new Figure 3d and accompanying .AVI video confirms this possibility. For clarity, we have recast the results on page 7 to state:

“...A closer examination of the CA1 region in the *Emx1^{Cre};R26^{P1}* and *Emx1^{Cre};R26^{P1R203W}* mice suggested PACS1 concentrates in the *stratum pyramidale* more than does PACS1^{R203W} (Fig. 3c). As the two HA-tagged proteins are expressed at similar levels in brain (Fig. 3a), this finding suggested the R203W substitution may cause PACS1 to distribute more broadly throughout the neuron.....

Reviewer #2 (Remarks to the Author):

This paper describes an RNAi based method to restore neuronal deficits in a PACS1 syndrome mouse model. Overall, the data presented is of high quality, but several important conclusions require additional experiments. The lack of an introduction and highly truncated discussion made this manuscript difficult to interpret. Major and minor concerns are listed below:

MAJOR

1) Significance for this work is drawn from the potential for novel therapeutics for PACS1 syndrome. However, no data shows that the mouse models created represent a reasonable model of the disease. Constitutive knockin of the mutation is lethal, which does not recapitulate the human condition. Given the poor translatability of animal research, a reasonable attempt should be made to show that the transgenic mice created model the disease, at least to some extent. This could be behavioral assays or measurements of neuronal activity in patient neurons, which could be obtained from direct-conversion of fibroblasts.

RESPONSE to comment 1: Please see our response to reviewer 1, comment 2 describing a new Figure 4d, which demonstrates that the Golgi deploys into the neurites of patient-derived NPCs, and that tubacin reverses this process. These data are very similar to those we report for the mouse models as shown in Figures 3d, 4c and S3c. Combined with our demonstration that *Pacs1^{KO}* mice are viable and display normal AP-independent release (Table S1 and Figure S6), these findings indeed suggest the mouse models we report here recapitulate the human disorder and identify HDAC6 and PACS1 as viable therapeutic targets to treat PACS1 syndrome patients.

Given that the R203W substitution has been reported in less than 150 patients worldwide, we were not surprised that constitutive expression of the disease protein is embryonic lethal in laboratory mice. We selected the *Emx1^{Cre}* driver because the scope of work in this paper focused on the effect of PACS1^{R203W} on neuronal structure and function in excitatory pyramidal neurons. However, its restricted

expression precludes use of this line in for meaningful behavioral assays. Those efforts are part of another study employing more broadly expressed Cre drivers.

2) Additional experiments are needed to bolster the major and central conclusion, which is that increased interactions between mutant PACS1 and HDAC6 drive PACS syndrome. The only evidence presented is co-immunoprecipitations from heterologous cells that overexpress proteins, which is insufficient. Patient cells present the perfect opportunity to test the hypothesis that R203W increases the association of PACS1 with HDAC6. It is surprising that coimmunoprecipitation and colocalization of endogenous proteins were not done since antibodies for endogenous proteins are available.

RESPONSE to comment 2: We must clarify that the singular result in Figure 2a demonstrating that the R203W substitution increases the interaction between epitope-tagged PACS1 and HDAC6 is certainly not the major and central conclusion of our work. Rather, it was the first clue that led to the identification of PACS1^{R203W}/HDAC6 as a key driver underpinning PACS1 syndrome. This identification is based on a wealth of cellular, genetic, enzymatic, anatomical, and electrophysiology studies employing patient fibroblasts, patient-derived NPCs, and three new mouse models, which all point to the importance of PACS1^{R203W}/HDAC6 in causing neuronal deficits. In every case, we tested these data using a combination of HDAC6 inhibitors, as well as siRNA and ASOs that target PACS1 or HDAC6 in each of the various patient and mouse models. We refer the reviewer to our opening response to reviewer 1 for a summary of the comprehensive and conclusive findings we report.

As requested by the reviewer, we performed a side-by-side colIP between endogenous HDAC6 and PACS1/PACS1^{R203W} in 160 (parent) and 159 (patient) fibroblasts heterozygous for PACS1^{R203W}, which suggest a modest increase in the 159 cells (Fig. S2c). We are not surprised at these results as we don't know the PACS1: PACS1^{R203W} stoichiometry in the 159 cells or to what extent the mutation may affect antibody avidity. Regardless, confocal analysis suggests a markedly increased interaction between PACS1/PACS1^{R203W} and HDAC6 in patient 159 cells (Fig. 2d), suggesting inherent limitations with the requested colIP experiment.

3) Given the focus on neuronal activity and known differences in golgi organization between cell types, mechanistic studies should also be done in neurons. While patient fibroblasts could be converted to neurons, an isogenic control should always be included to control for the differences in genetic architectures, even among family members.

RESPONSE to comment 3: Please see our responses to Reviewer #1, comments 2 and 3.

4) Measurements of Golgi localization should be controlled for the size of the cell and polarity of the golgi stack. Patient cells seem to be larger (Fig1,S1), so one could expect that their perinuclear distance is also affected given the absolute measurements described in the methods. Given the proposed disruptions to microtubule acetylation, one would expect changes in cellular morphology.

RESPONSE to comment 4: The patient fibroblasts are indeed frequently larger, with a complex and irregular cytoplasm as indicated by the cell tracings in Figure 1a. We revised the text on page 4 to note this difference. Golgi positioning in fibroblasts, however, is dependent on positioning of the single primary MTOC. Thus, even a 4hr treatment of the patient cells with the HDAC6 inhibitor tubacin rescues Golgi positioning without affecting the overall cell size (see Figure 2c). This difference extends to dissociated DIV5 hippocampal neurons. In R26^{P1} neurons, the Golgi remains concentrated in the cell

soma, despite a markedly complex and elaborate cytosol. In R26^{P1R203W} neurons, however, the Golgi fragments and disperses down every neurite (Fig. 3d and Movie S1). Our studies suggest HDAC6 disturbs MTOC hierarchy as evidenced by the nocodazole washout studies, which demonstrate a single MTOC in parent 160 cells but multiple MTOCs in patient 159 cells (Figure S1b).

5) The electrophysiology results are difficult to reconcile with the model. The authors argue that ultimately, HDAC6-dependent acetylation alters pre and postsynaptic function. However, HDAC6 knockdown does not alter neuronal morphology (Fig 2F) or synaptic activity (Fig 3a). Moreover, the PACS1 KO mice show elevated Acetyl-tubulin (Fig S5d), but do not show synaptic deficits. These results undermine the central hypothesis.

RESPONSE to comment 5: Contrary to this referee's premise, our electrophysiology studies on the PACS1 syndrome mice and PACS1^{KO} mice are both internally consistent and in full agreement with the scientific literature. Specifically, our result demonstrating the lack of effect of H6ASO on mEPSCs in R26^{P1} mice (Figure 5e), are in full agreement with the report by Lee et al 2012 (Ref. 33), which demonstrated Hdac6 knockdown in mPFC has no effect on mEPSCs. We have recast the results on pages 8 and 9 to state:

“...Importantly, the H6ASO treatment in R26^{P1R203W} mice fully restored both mEPSC amplitude and frequency to levels observed in R26^{P1} mice, which were unaffected by HDAC6 depletion (Fig. 5e). These findings suggest PACS1^{R203W}/HDAC6 disturbs basal glutamatergic transmission and, in agreement with others, that HDAC6 knockdown itself in mPFC has no deleterious effect on mPSCs³³...”

Similarly, our results that the suppressed mEPSC frequency and amplitude in the R26^{P1R203W} mice (Figure 5e) and Pacs1^{+R201W} mice (Figure 6a) are restored with H6ASO or P1ASO, respectively, are in full agreement with studies by Xing et al 2020 (reference 49) demonstrating that Dip2b knockdown, which also reduces acetylated α -tubulin, suppresses mEPSC frequency, and with the report by Ma et al 2019 (reference 31), demonstrating Dip2a knockdown, which also reduces acetylated cortactin, suppresses mEPSC amplitude. We discuss these findings on page 12.

It is also important to point out that unlike the muted effect of the H6ASO knockdown or Pacs1^{KO} (both of which increase acetylated α -tubulin as shown in Figures 4c and S5c and d), it is the loss of the tubulin acetyltransferase (which decreases acetylated α -tubulin) that has profound effects on neurons. For example, in worms MEC-17 KO causes axonal degeneration (PMID:24373971). In mice, α TAT deletion impairs neuronal migration leading to septal and striatal hypoplasia and dilation of the lateral ventricles (PMID:30953095). Together, these studies further support our electrophysiology results and suggest hypoacetylation of α -tubulin is much more problematic for neuronal function than hyperacetylation.

MINOR/SPECIFICS

1) At least one other golgi marker should be used to show that PACS1 is not altering the localization of Giantin, which is typically localized to cis-Golgi.

RESPONSE to minor comment 1: In a new Figure S1a, we used anti-GM130 to stain the Golgi in patient 650 and parent 651 cells and observed no difference in Golgi positioning compared to the results using anti-Giantin (Figure 1a). In a new Figure 4c, we use anti-GM130 to stain the Golgi in the R26^{P1} and R26^{P1R203W} hippocampus and found similar results as we describe using anti-Giantin in the DIV5 hippocampal neurons shown in Figures 3d and S3c. Please also see our response to reviewer #1, comment 2.

2) Authors should determine and quantify whether wt or mutant PACS1 alter HDAC6 overall expression. Some figures seem to suggest this (Fig1F), which would significantly change the interpretation of results (Fig 1J).

RESPONSE to minor comment 2: We have requantified our data and find no consistent effect of PACS1 expression on HDAC6 levels.

3) Authors state that there is more PACS1R203W in dendrites (Fig 2c). This reviewer doesn't see this, so it should be quantified.

RESPONSE to minor comment 3: Please see our response to reviewer #1, minor comment 2.

4) In general, immunocytochemistry of neuronal cultures is poor (Fig 2d). Golgi structures should be clearly visible and do not normally fill the cytoplasm as shown. PACS1 also does not seem to localize to Golgi as expected. Authors report golgi fragments in dendrites, but these are almost certainly axons. This would be odd as neuronal axons should not have golgi structures.

RESPONSE to minor comment 4: Please see our response to Reviewer #1, comment 1.

5) Reconstructions of neuronal morphology from Tdtomato-filled AAV-infect neurons is impressive. Images should be included as this reviewer does not understand how this could be done without sparse labeling of neurons (typically done with a different virus).

RESPONSE to minor comment 5: Please see our response to Reviewer #1, comment #2.

6) Synaptosomal fractions are notoriously contaminated. The protein expression patterns shown in Fig S4a (decreasing amount) are typically interpreted as being absent from synapses.

RESPONSE to minor comment 6: We believe our synaptosome fractionation data are sound and similar quality as those published by others (e.g. PMID: 29610302). Like others have reported, AMPAR1, GABAaRa1, and PSD95 are each quantitatively found in the synaptosomal fraction, which reflects their singular functions (Fig. S5a). By contrast, multifunctional PACS1, which is present on endosomes (light membranes) and in the cytosol (see reference 7, Figure 2b, and reviewed in reference 6), also localizes to synaptosomes, so its partitioning would not be expected to be "binary". Our determination that PACS1 and HDAC6 are present in the synaptosome fraction is consistent with confocal analysis demonstrating PACS1 and HDAC6 colocalize in dendritic spines (Fig. 5c). Our data are also supported by the report by Peixoto et al 2019 (PMID 31722214), which independently demonstrated PACS1 is present in dendritic spines.

7) 5-10 hz frequencies for Minis is very high. An explanation should be provided for these results.

RESPONSE to minor comment 7: The frequencies for the minis reported here are comparable to those reported by others. For example, Bridi et al 2020 (Reference 51) reported mPSCs in L2/3 neurons from 5-10 week old mice with frequencies ~9-10 Hz and Antione (Reference 50) reported mPSC frequencies in the range of ~12-25 Hz in L2/3 neurons from P17-P23 mice.

8) The liquid phase separation data is superficial and does not contribute to the main hypothesis.

RESPONSE to minor comment 8: We deleted LLPS connection and the G3BP1 data, and moved the p62 body data to Figure 2d, which supports the increased interaction between endogenous PACS1^{R203W} and HDAC6.

Reviewer #3 (Remarks to the Author):

This is a comprehensive translational study that elucidates further the cellular defects underlying PACS1 syndrome; reports new mouse models of the recurrent PACS1-R203W causal variant; and tests pharmacological and ASO-based therapies on neuronal structure and synaptic transmission in vivo. Using patient and parent fibroblasts, the authors show that PACS1-R203W disrupts Golgi positioning and microtubule organization, offers functional links to α -tubulin acetylation and show that R203W increases the interaction with HDAC6. Pharmacological and genetic HDAC6 inhibition were able to rescue these cellular defects significantly, prompting validation in vivo. Using a newly generated mouse model of PACS1 syndrome with a humanized conditional allele bearing PACS1-R203W in the Rosa26 safe harbor locus, the authors expressed WT or variant PACS1 in Emx1-Cre-driven neurons. The Golgi positioning and dendrite morphology, as well as synaptic function, were also altered in PACS1-R203W mouse models, defects that could be rescued significantly with HDAC6 ASOs. Finally, the authors pursued the possibility of targeting PACS1-R203W directly with ASOs through generation of Pacs1 KO and conditional Pacs1-R201W KI mouse models. Overall, the manuscript is original, represents a substantial amount of work to advance our knowledge of the molecular basis of PACS1 syndrome, highlights the PACS1-HDAC6 interaction as a key contributor and to pathology, and excitingly, shows that the syndrome may be treatable with either HDAC inhibitors or ASOs. The methodology is well described and the data are beautifully presented in each figure.

I am enthusiastic about this work advancing toward publication and have only a few comments.

1. Please show dot plots on all charts, or at least overlay them onto the bars. This is particularly important for the n=3 datasets.

RESPONSE to comment 1: As requested, we show dot plots in all figures in the revised manuscript.

2. Figure 2c shows differential subcellular localization of HA-tagged PACS1 and PACS1-R203W. First, it is really difficult to see this difference given the bright signal throughout the section so please modify the presentation of the image to enable the reader to more readily see this difference. Additionally, please show how the ha-tag of the transgene compares to localization of endogenous WT PACS1 with a co-stain.

RESPONSE to comment 2: Please see our response to Reviewer #1, minor comment 2. Please note that we lack antibodies specific for human PACS1, preventing us from comparing the distribution of the mouse and HA-tagged human proteins.

3. Please quantify the protein levels in Figure 4b to better characterize the impact on HDAC6 and WDR37.

RESPONSE to comment 3: As requested, we include quantitation of PACS1, PACS2, HDAC6 and WDR37 in Figure 6b (old Figure 4b).

4. The notable rescue of Golgi positioning with not one, but three different HDAC inhibitors as shown in Figure S2 is underplayed in the main text and should be emphasized further.

RESPONSE to comment 4: As requested by the referee, we now show the effects of tubacin versus AGK2 on Golgi positioning in patient cells in revised Figure 2c.

5. Although this is already an extensive dataset, one piece that would really complement this story is to know whether PACS1-R203 or Pacs1-R201 mouse models have been characterized through behavioral testing. While the structural and functional phenotype readouts are rational measures for therapeutic amelioration with HDAC6 inhibitors or ASOs, this would offer robust in vivo data to support further the authors' conclusions.

RESPONSE to comment 5: Please see our response to referee 2, comment 1.

Reviewers' Comments:

Reviewer #1:

Remarks to the Author:

In this revised version of the manuscript by Villar-Pazos and colleagues, the authors used a combination of molecular and cellular analyses on patient-derived cells in vitro and transgenic mice to study the effects of a missense mutation in the PACS1 gene, which encodes a multi-functional trafficking protein and is linked to a neurodevelopmental genetic disorder characterized by intellectual disability.

The authors have performed additional experiments and incorporated changes to the figures and text that respond to all my criticisms. I also noticed that they resolved most issues raised by the other reviewers. I consider this manuscript to be in a format that is ready for publication.

I have two additional suggestions:

- now that dots have been added to all bar graphs to represent individual measurements, the figure legends should indicate which type of replicate is involved. For example, in WB experiments in Figures 1c and 1e, is $n=3$ and $n=5$ the number of independent experiments or the number of samples included in one single experiment (which would be preferable)? This impacts the way variability is represented in the control bars in these panels (160, 651, and WT): if the n is the number of samples in one single experiment, then the mean band intensity should be normalized to 1, not each individual control measurement. In this case, the control bars would still indicate variability (SEM, for instance), which should be taken into account in all subsequent statistical testing. The same information should be included in the Reporting Summary and in the Source Data file. Similar instances exist in Figures 2a, 2b, 2g, and 5d. I also notice that the best practice for reporting inter-replicate variability has been used in figure 6c, which should be used as a reference by the team. On the other hand, if those replicates are independent experiments, this should be indicated in the legend and reporting summary, with a note that normalization was performed across samples in each individual blot.

- I am still not happy with the title. If the authors used patient-derived cells and transgenic mice, why is the title focusing on the RNA-targeted therapy experiments conducted in mice? I suggest the authors find a better way to succinctly convey all of what they have done in a short title.

Reviewer #2:

Remarks to the Author:

The authors have made revisions that address some concerns, but others remain.

1- An important concern that this story was presented in "bits and pieces", mostly remains. Additional cell models introduced now means that data is presented in patient fibroblasts, immortalized HeLa cells, MEFs, NPCs, primary neuronal cultures, and neurons in slices. Evidence that all major conclusions can be observed in at least one type of cell (ideally neurons) would provide rigor. The cell types used have unique cell biologies and the argument that together they present strong evidence to some effect is simply not compelling. For example, unlike heterologous cells, bona fide neurons have many microtubule organizing centers (a focus of some of their experiments).

2- Two reviewers expressed concerns about the quality of Fig 3D, yet the authors chose to analyze the same image and in greater detail using Bitplane (Imaris). This is inappropriate. The resolution and immunostaining of this image is relatively poor, which reduces confidence of the quantified data.

3- Multiple conclusive statements are made throughout the manuscript that should be either quantified by data, toned down, or removed. These include: 1) patient fibroblasts are larger, 2) more irregular, 3) proliferate more slowly, 4) have abnormal microtubule organization. 5) HeLa cells have disorganized MTs. 6) HDAC ASOs rescue Golgi positioning in neurons in hippocampal slices (fig 4C). 7) There is increased colocalization between PACS1 mutant and HDAC6 (Fig 2d). 8) PACS1 patient NPCs

precociously generated neurite-like processes. There are others. Cell size is a confound since organelle size can vary with cell size (see Wallace F. Marshall, 2020, Annual Reviews of Cell and Developmental Biology). Measurements of Golgi structures should be normalized to cell size.

4- New fig 4C presents high quality images of Golgi structures in bona fide neurons "in vivo" and conclusions are made of their altered location. However, this was not quantified. Rescues of Golgi positioning by ASOs alluded to in the text are neither shown nor quantified. It is important to note that Golgi structures are normally found in primary dendrites.

5- The authors performed IPs of endogenous proteins, which is good, but do not find increased association between the mutant and HDAC6. However, they dismiss these findings and state that there is increased association of endogenous proteins as observed by colocalization in images (2d). However, as with other statements, this is not quantified.

6- The term "Knock in" is reserved for changes to the endogenous locus, yet their mouse model is an overexpressed tagged transgene. This should be corrected as it can be misinterpreted.

7- Several statements need to be amended- "these findings indeed suggest the mouse models we report here recapitulate the human disorder". No evidence presented suggests this is true. Moreover, the discussion should focus on the inconsistent data- For example, that Golgi positioning is strongly dependent on acetylation, but that the PACS1 KO had no effect on Golgi positioning.

Reviewer #3:

Remarks to the Author:

I remain highly enthusiastic about this manuscript and found the authors to be attentive to the comments from all three reviewers. The new data and improved presentation strengthen the overall message. I would endorse publication if the following minor comments are addressed:

1. I appreciate the response to my previous comment requesting quantification of immunoblots. The numerical densitometry values for normalized bands overlaid on the blot image is not very reader friendly and would be better suited to a graph to show differences. The indication of just a single value in such charts (i.e. Fig 6b) suggests that the experiment was performed just once, which is problematic and begs the question of whether IB results were reproducible. Please comment on biological replicates, or were samples from multiple animals pooled?

2. The revised article format of the manuscript with subheadings is helpful toward leading the reader through each aspect of the overall body of work. However, this makes the last section on the PACS1-PACS2-WDR37 regulatory network appear abruptly and briefly. An improved transition here would be helpful. Further, if WDR37 and PACS2 are such critical players in this network, then please incorporate them into Fig. 6d.

REVIEWER COMMENTS

Reviewer #1 (Remarks to the Author):

In this revised version of the manuscript by Villar-Pazos and colleagues, the authors used a combination of molecular and cellular analyses on patient-derived cells in vitro and transgenic mice to study the effects of a missense mutation in the PACS1 gene, which encodes a multi-functional trafficking protein and is linked to a neurodevelopmental genetic disorder characterized by intellectual disability.

The authors have performed additional experiments and incorporated changes to the figures and text that respond to all my criticisms. I also noticed that they resolved most issues raised by the other reviewers. I consider this manuscript to be in a format that is ready for publication.

I have two additional suggestions:

- now that dots have been added to all bar graphs to represent individual measurements, the figure legends should indicate which type of replicate is involved. For example, in WB experiments in Figures 1c and 1e, is n=3 and n=5 the number of independent experiments or the number of samples included in one single experiment (which would be preferable)? This impacts the way variability is represented in the control bars in these panels (160, 651, and WT): if the n is the number of samples in one single experiment, then the mean band intensity should be normalized to 1, not each individual control measurement. In this case, the control bars would still indicate variability (SEM, for instance), which should be taken into account in all subsequent statistical testing. The same information should be included in the Reporting Summary and in the Source Data file. Similar instances exist in Figures 2a, 2b, 2g, and 5d. I also notice that the best practice for reporting inter-replicate variability has been used in figure 6c, which should be used as a reference by the team. On the other hand, if those replicates are independent experiments, this should be indicated in the legend and reporting summary, with a note that normalization was performed across samples in each individual blot.

>> We normalized the band densitometry in figures 1c, 1e, 2a, 2b, 2g, and 5d to the WT sample and the appropriate loading controls separately to minimize the contribution of inter-experiment variability characteristic of western blot signals. We recast the respective figure legends on pages 29, 30, and 33 to clarify this point. We made similar edits for western blots in Figures S2c (page 40) and a new S7d (page 45, per referee #3, comment 1).

- I am still not happy with the title. If the authors used patient-derived cells and transgenic mice, why is the title focusing on the RNA-targeted therapy experiments conducted in mice? I suggest the authors find a better way to succinctly convey all of what they have done in a short title.

>> As requested, we recast the title to deemphasize the mouse ASO studies and better convey to scope and depth of this study. The new title is: **“Neural deficits in PACS1 syndrome caused by dysregulated HDAC6 are corrected with targeted therapy”**. Accordingly, we made minor edits to the Abstract (shown in text-edit mode) to better reflect the scope of the new title.

Reviewer #2 (Remarks to the Author):

The authors have made revisions that address some concerns, but others remain.

1- An important concern that this story was presented in “bits and pieces”, mostly remains. Additional cell models introduced now means that data is presented in patient fibroblasts, immortalized HeLa cells, MEFs, NPCs, primary neuronal cultures, and neurons in slices. Evidence that all major conclusions can be observed in at least one type of cell (ideally neurons) would provide rigor. The cell types used have unique cell biologies and the argument that together they present strong evidence to some effect is simply not compelling. For example, unlike heterologous cells, bona fide neurons have many microtubule organizing centers (a focus of some of their experiments).

>> The isogenic Hela cell data (Figure 1b) and patient NPC data (Figure 4d) were included in the revised manuscript in response to referee #1 (comments 2 and 3), and referee #2 (comments 1 and 3). We agree that neural MTOCs include key roles for Golgi outposts. Indeed, our findings suggest the increased number of Golgi outposts in PACS1^{R203W} neurons contributes to the increased dendritic complexity in PACS1 syndrome. We discuss this point on page 11 where we state:

“The reversible acetylation of α -tubulin is essential for MT nucleation at centrosomes (acetylated) and for driving transition of the Golgi ribbon to dispersed mini-stacks (deacetylated), a step critical for dendritic growth and branching⁴¹. In flies, the Golgi does not present as a ribbon but is maintained as dispersed elements called “outposts”, which efficiently sort into the growing dendrites²³. The deployed Golgi elements localize to branch points where they support local secretory pathway trafficking needed by the growing arbor and may also function as secondary MTOCs^{20,22,23}. Compared to flies, the characteristic Golgi ribbon in mammalian cells has been suggested to explain the relative infrequency of dendritic Golgi observed in mammalian neurons²⁰. Nonetheless, in mammalian cortical and hippocampal pyramidal neurons, the Golgi and associated secretory cargo flux are polarized toward long primary dendrites, but not axons. Thus, the markedly elevated number of Golgi fragments dispersed into the dendrites in PACS1^{R203W} neurons correlate with the increased neuronal arborization.” ...

2- Two reviewers expressed concerns about the quality of Fig 3D, yet the authors chose to analyze the same image and in greater detail using Bitplane (Imaris). This is inappropriate. The resolution and immunostaining of this image is relatively poor, which reduces confidence of the quantified data.

>> We used high-resolution confocal microscopy combined with tiling (Nikon Elements) to capture the images of the hippocampal neurons shown in Figures 3d and S3c. We recast the text in Methods and in the figure legends to clarify this point. Specifically, in Methods on page 20, we state:

...“High-resolution tiled images of entire cultured hippocampal neurons were captured using a Nikon Ti2-E confocal microscope equipped with high-sensitivity GaAsP detectors, and with a 60x oil immersion objective and 3x zoom (pixel size (XY) = 0.069 μ m/pixel, Z step = 0.15 μ m).” ...

This method enabled us to use these images for both the high-magnification Imaris reconstructions shown in Figure 3d (page 31), and the low-magnification maximum intensity projections of the whole neurons shown in Figure S3c (page 41).

3- Multiple conclusive statements are made throughout the manuscript that should be either quantified by data, toned down, or removed. These include: 1) patient fibroblasts are larger, 2) more irregular, 3) proliferate more slowly, 4) have abnormal microtubule organization. 5) Hela cells have disorganized MTs. 6) HDAC ASOs rescue Golgi positioning in neurons in hippocampal slices (fig 4C). 7) There is increased colocalization between PCAS mutant and HDAC6 (Fig 2d). 8) PACS1 patient NPCs precociously generated neurite-like processes. There are others. Cell size is a confound since organelle size can vary with cell size (see Wallace F. Marshall, 2020, Annual Reviews of Cell and Developmental Biology). Measurements of Golgi structures should be normalized to cell size.

>> Regarding items 1-3, in the original review, this referee correctly suggested the patient cells seemed to be larger. In response to this referee's comment, we added this observational point to page 4 of the revised manuscript. We have elected not to change this text as we believe it is instructive and relates the cell images in Figure 1 to the quantitative analysis of Golgi localization. Please also see our response to this referee's comment on Golgi positioning below.

>> Regarding items 4 and 5, we believe that the quantitative microtubule aster assay shown in Figure S1b, which demonstrates patient cells are beset with supernumerary MTOCs, supports our statement that MT organization is disturbed in patient cells.

>> Regarding item 6, referee #1 (comment 2) had requested that we measure the effect of the ASOs on Golgi positioning in the brain specimens to determine if the increased dendrite complexity of the R26^{P1R203W} hippocampus was coupled to an increase in dendritic Golgi. Accordingly, we stained 50 μ m slices of the TdTomato⁺ specimens for GM130. This analysis was technically constrained by the requirement to assess i) GM130 staining only within the short segments of the TdTomato⁺ apical dendrite included in the 50 μ m slices, and ii) only those TdTomato⁺ neurons in which the GM130 staining was separated from Golgi elements in adjacent cells (see Figure 4c). These limitations permitted only a qualitative assessment of Golgi positioning. Nonetheless, as we show in Figure 4c, R26^{P1R203W}/nASO neurons harbor Golgi elements that extended farther along the apical dendrite than R26^{P1R203W}/H6ASO specimens. These findings correlated nicely with the quantitative analysis of Golgi positioning in the isolated hippocampal neurons shown in Figure 3d. To satisfy this referee, we recast the text on pages 7 and 8 to state:

..." Consistent with our quantitative studies in the dissociated hippocampal neurons (**Figs. S3c and 3d**), confocal imaging suggested that R26^{P1R203W} neurons treated with nASO harbored Golgi elements that deployed into the TdTomato⁺ apical dendrite (**Fig. 4c**). By contrast, treatment with H6ASO caused the Golgi to return to the cell body. As expected, neither ASO affected the localization of the Golgi to the cell body in the R26^{P1} neurons."

>> Regarding item 7, in the re-revised manuscript, we include quantitation (Pearson's coefficient) of the increased co-localization of HDAC6 with PACS1/PACS1^{R203W} in patient cells. Specifically, in Results on page 6, we state:

..."Image analysis revealed bodies positive for HDAC6 and PACS1/PACS1^{R203W} were observed in 39% of patient cells (Pearson's coefficient = 0.55) but only 2% of parental control cells (Pearson's coefficient = 0.15), suggesting PACS1^{R203W} also impacts HDAC6-mediated proteostasis pathways (**Fig. 2d** and see Discussion)."...

And in Methods on pages 19 and 20, we state:

..." Colocalization of PACS1/PACS1^{R203W} with HDAC6 was quantified through Pearson's coefficient (R) using the Coloc2 plugin integrated in Fiji (ImageJ). Deconvolved z-stack images were segmented to identify high-density bodies based on the intensity of the fluorescent signals, the object area (> 2 μm^2), and their circularity. The bodies were then analyzed for the spatial correlation (colocalization) between PACS1/PACS1^{R203W} and HDAC6."...

>> Regarding item 8, the precocial generation of neurites in the patient NPCs, which invariably occurred on day 18 of the differentiation protocol, is important and needs to be stated. To satisfy this referee, we recast the sentence on page 8 to state:

..." Interestingly, on day 18 of the neural induction protocol, the PACS1^{R203W} NPCs precociously generated neurite-like processes that were not yet apparent in the parental NPCs (**Fig. 4d**). In addition, and similar to the R26^{P1R203W} pyramidal neurons, the neurites in the patient-derived NPCs contained Golgi elements that disseminated from the cell body into the neurite. A 24-hr treatment with the HDAC6 inhibitor tubacin reversed this effect, restoring Golgi localization to the paranuclear region. These findings suggest the effects of PACS1^{R203W}/HDAC6 on neuronal structure in mice recapitulate important aspects of the human disorder."...

>> Regarding the analysis of Golgi positioning, the Marshall review (PMID 32603615) focuses on the scaling of subcellular structures, notably nuclei, and flagella, and makes a point to state that scaling of secretory pathway compartments is autonomously controlled by curvature-inducing proteins. Indeed, our automated analysis of Golgi positioning and fragmentation controlled for changes in nuclear size and shape. To clarify this point and satisfy this referee, we include a detailed description of the Golgi morphometry methods on page 19 that states:

..."Golgi localization (juxtannuclear vs peripheral) and fragmentation analyses were designed and automated using Nikon Elements software. Deconvolved multi-channel images were segmented using automatic Otsu thresholding. An ROI dividing the cytosolic compartments was automatically generated in each cell with a perimeter set 10 μm (dermal fibroblasts) or 3 μm (MEFs, HeLa cells, and NPCs) from the edge of the nuclear mask. The juxtannuclear/peripheral border was automatically determined and dependent on the shape and size of the nucleus of each cell. Golgi dispersion was determined by the position of Golgi objects (percentage of total Golgi area) relative to the ROI. Juxtannuclear Golgi included all Golgi objects located between the margin of the ROI and the nuclear envelope. Peripheral Golgi included all objects outside the ROI border."...

4- New fig 4C presents high quality images of Golgi structures in bona fide neurons “in vivo” and conclusions are made of their altered location. However, this was not quantified. Rescues of Golgi positioning by ASOs alluded to in the text are neither shown nor quantified. It is important to note that Golgi structures are normally found in primary dendrites.

>> Please see our response to query 1 and also query 3, item 6.

5- The authors performed IPs of endogenous proteins, which is good, but do not find increased association between the mutant and HDAC6. However, they dismiss these findings and state that there is increased association of endogenous proteins as observed by colocalization in images (2d). However, as with other statements, this is not quantified.

>> Please see our response to query 3, item 7.

6- The term “Knock in” is reserved for changes to the endogenous locus, yet their mouse model is an overexpressed tagged transgene. This should be corrected as it can be misinterpreted.

>>We disagree. The term “knock-in” to describe the introduction of cDNAs into the ROSA26 locus is broadly accepted by animal model geneticists (e.g., see PMIDs 26772810, 36441701, and 27063570). Rather, the term “overexpressed transgene” is problematic as it could be misinterpreted as merely a comparison of mouse lines harboring random transgene integrations.

7- Several statements need to be amended- “these findings indeed suggest the mouse models we report here recapitulate the human disorder “. No evidence presented suggests this is true.

>> We disagree. Figures 3d, 4c, and 4d show that PACS1^{R203W} causes an excessive redistribution of Golgi elements into the processes of mouse and human neural cells, and silencing of HDAC6 expression or activity reverses this effect. Thus, our findings indeed suggest that the mouse models we report here recapitulate the human disorder. Please also see our response to query 3, item 8. To satisfy this referee, we recast the text on page 8 to state:

...“These findings suggest the effects of PACS1^{R203W}/HDAC6 on neuronal structure in mice recapitulate important aspects of the human disorder.”...

Moreover, the discussion should focus on the inconsistent data- For example, that Golgi positioning is strongly dependent on acetylation, but that the PACS1 KO had no effect on Golgi positioning.

>> We disagree and we elaborated on this point in response to the initial review. Briefly, *hyperacetylation* of α -tubulin, which occurs in PACS1 KO MEFs, has no effect on Golgi positioning (see Figure 1f). By contrast, *hypoacetylation* of α -tubulin, which occurs in cells expressing PACS1^{R203W}, markedly disturbs Golgi positioning in cell lines, mouse neurons, and NPCs (Figures 1, 3 and 4). Indeed, the ability of K⁴⁰Q- α -tubulin to rescue Golgi positioning in patient cells supports these findings (Figure 1d).

Reviewer #3 (Remarks to the Author):

I remain highly enthusiastic about this manuscript and found the authors to be attentive to the comments from all three reviewers. The new data and improved presentation strengthen the overall message. I would endorse publication if the following minor comments are addressed:

1. I appreciate the response to my previous comment requesting quantification of immunoblots. The numerical densitometry values for normalized bands overlaid on the blot image is not very reader friendly and would be better suited to a graph to show differences. The indication of just a single value in such charts (i.e. Fig 6b) suggests that the experiment was performed just once, which is problematic and begs the question of whether IB results were reproducible. Please comment on biological replicates, or were samples from multiple animals pooled?

>> The western blot data in Figure 6b were analyzed in three independent experiments. We recast the legend on page 34 to clarify this point where we state:

..."(b) Western blots of cortical lysates from mice in panel a. Numerical values depict the normalized mean signal intensity of the PACS1, PACS2, HDAC6, and WDR37 bands, n = 4 mice in three independent experiments (graphs shown in Figure S7d)."...

We include the statistical analysis in a new Figure S7d, demonstrating the PACS1 ASO-induced reductions in PACS1/PACS1^{R201W}, PACS2 and WDR37, and the PACS1^{R201W}-induced reduction of PACS2 are statistically significant. Accordingly, in the legend to Figure S7d on page 45, we state:

..."Quantification of western blot signals for PACS1, PACS2, HDAC6, and WDR37 in the cortical lysates shown in Figure 6b. Data are mean ± SEM (2-way ANOVA followed by LSD *post hoc* test), n = 3 independent experiments, normalized individually to minimize inter-experimental variability."...

2. The revised article format of the manuscript with subheadings is helpful toward leading the reader through each aspect of the overall body of work. However, this makes the last section on the PACS1-PACS2-WDR37 regulatory network appear abruptly and briefly. An improved transition here would be helpful.

>> We agree and have removed the section subheading.

Further, if WDR37 and PACS2 are such critical players in this network, then please incorporate them into Fig. 6d.

>> The literature lacks sufficient information to include WDR37 or PACS2 in the steps depicted in Figure 6d. Therefore, we recast the legend to Figure 6 on page 33 to state:

..." Potential roles for PACS2 or WDR37 in these processes are not considered."....

Reviewers' Comments:

Reviewer #2:

Remarks to the Author:

-The authors have addressed some concerns, but some fundamental concerns remain. The most important of which is the development of a predominantly cell-biology story distributed throughout many different cell types. This means that conclusions are difficult to assess, which compromises the impact of results

-The important point regarding the use of the term "knockin" was to make clear that this is not a mutation of the endogenous protein, and that the protein with the disease-related mutation is overexpressed in their system. This is a crucial component of this work that is not made clear.

REVIEWERS COMMENTS

Reviewer #2 (Remarks to the Author):

-The authors have addressed some concerns, but some fundamental concerns remain. The most important of which is the development of a predominantly cell-biology story distributed throughout many different cell types. This means that conclusions are difficult to assess, which compromises the impact of results

Response: We view the analysis of multiple cellular and mouse models as a strength of this work since the results obtained with the complementary models are fully consistent and, together, strongly suggest PACS1^{R203W}/HDAC6 underpins critical neural deficits associated with PACS1 syndrome. It is important to note that our study began with patient fibroblasts, the only available tool at the time. The cultured cell studies readily suggested that R203W is gain-of-function disease substitution that aberrantly potentiates HDAC6 activity to disturb Golgi positioning and microtubule organization. As we developed and interrogated additional cellular and mouse models of PACS1 syndrome, we confirmed the importance of PACS1^{R203W}/HDAC6 to the Golgi-associated neural deficits associated with PACS1 syndrome. Furthermore, we applied numerous strategies (siRNAs, enzyme inhibitors, and ASOs) to the various models to validate our findings, which suggest this neurodevelopmental disorder may be treatable with targeted therapies. Thus, using multiple models we were able to decipher the mechanism underlying this devastating disorder and how it may be treated, underscoring the significance of our study.

-The important point regarding the use of the term "knockin" was to make clear that this is not a mutation of the endogenous protein, and that the protein with the disease-related mutation is overexpressed in their system. This is a crucial component of this work that is not made clear.

Response: We believe we were careful to distinguish the use of CRISPR/Cas9 gene editing to knock-in Cre-inducible, HA-tagged human PACS1 or PACS1^{R203W} into the murine ROSA26 safe harbor locus (Page 7) yielding the matched R26^{P1} and R26^{P1R203W} lines from the use of CRISPR/Cas9 to conditionally express the knocked-in disease mutation from the *endogenous Pacs1* locus, yielding the *Pacs1*^{R201W/+} mice. As the referee suggests, knowing how much HA-tagged PACS1 or PACS1^{R203W} is expressed in the Rosa26 model is crucial. Indeed, on page 7, we state that “Western blot and immunohistochemical staining showed *Emx1Cre* faithfully induced expression of HA-tagged PACS1 or PACS1^{R203W} at levels similar to endogenous PACS1 in the cortex and hippocampus (Figs. 3a and b).” Importantly, we demonstrated that the dendritic overbranching observed in the R26^{P1R203W} mice recapitulated the precociously generated neurite-like processes we observed in patient NPCs (Fig. 4d), underscoring the value of the R26 strategy. In addition, our electrophysiology studies showed similarly reduced mEPSCs in the prefrontal cortex of both the R26^{P1R203W} mice and the *Pacs1*^{R201W/+} mice, further suggesting the validity of the Rosa26 strategy described in our study. Finally, we also controlled for the megamer strategy used to generate the *Pacs1*^{R201W/+} mice by producing the *Pacs1*^{M/+} control mice. Gratifyingly, we found the megamer had no effect on our findings. In summary, we believe that we clearly distinguished knock-in strategies used to generate the R26^{P1}/R26^{P1R203W} and the *Pacs1*^{M/+}/*Pacs1*^{R201W/+} lines and that we showed the two models yielded consistent and complimentary results.